# Efficient Distributed Principal Component Analysis with Parallel Deflation

## Abstract

We study a distributed Principal Component Analysis (PCA) framework where each worker targets a distinct eigenvector and refines its solution by updating from intermediate solutions provided by peers deemed as "superior". Drawing intuition from the delation methods, which is traditionally used in centralized eigenvalue problems, our method breaks the sequential dependency in between the deflation steps and allows asynchronous updates of workers while incurring only a small communication cost. To our knowledge, a critical gap in the literature –*the theoretical underpinning of such distributed, dynamic interactions among workers*– has remained unaddressed until now. This paper offers the first theoretical analysis explaining why, how, and when these intermediate, hierarchical updates lead to practical and provable convergence in distributed environments. Our theoretical contributions demonstrate that such a distributed PCA algorithm not only converges effectively but does so in a manner that is favorably scalable. We also demonstrate through experiments that our proposed framework offers comparable performance to EigenGame-$\mu$, the state-of-the-art model-parallel PCA solver.

## 1 Introduction

Currently, datasets have gotten dramatically large, encompassing billions, if not trillions, of entries spanning various domains Zhong et al. (2019); Liu et al. (2023); Penedo et al. (2024a); Soldaini et al. (2024); Wang et al. (2022); Schuhmann et al. (2022); Raffel et al. (2020); Penedo et al. (2024b); Xue et al. (2020); Abadji et al. (2022). This scale made it necessary to advance various distributed optimization protocols, such as federated learning Brendan et al. (2016), and, notably, the development of multiple distributed ML software packages Kim & Kang; Dean et al. (2012). Specialized frameworks such as Ray Liang et al. (2018), Spark Meng et al. (2016), Hadoop Apache Software Foundation, and JAX Bradbury et al. (2018) have surged in popularity due to their ability to enhance computational speed significantly.

However, at the algorithmic level, most distributed implementations try to simulate the behavior of the centralized versions of the underlying algorithms. That is, how distributed algorithms navigate the parameter landscape is often designed such that we achieve a similar outcome as if data is available in one location. There are a few key reasons for this:

- **Mathematical Understanding**: When there is sufficient theoretical understanding of the centralized version, it is often a desired goal to attain the same result by designing algorithms to emulate the centralized counterparts. This ensures consistency and theoretical understanding.
- **Algorithm Simplicity**: Since centralized algorithms are better understood, distributed variants that replicate the algorithms' outcomes automatically enjoy the same simplicity and interpretation.
- **Benchmarking**: By simulating the centralized execution, comparing the accuracy and convergence properties of the distributed implementation in practice becomes easier.

Yet, precisely simulating centralized algorithms in a distributed environment could pose some challenges. Take as a characteristic feature the notion of *synchrony* in distributed implementations, as this leads to training dynamics closer to centralized training. Synchronization among workers means proper orchestration, especially in large-scale settings with high-dimensional models and datasets Zeng et al. (2024); Tan et al. (2022). Synchronized implementations that wait for some or all workers to finish each iteration before proceeding can suffer from stragglers and load imbalance Wang et al.

(2023a); Ambati et al. (2019). Yet, while asynchronous motions seem like a favorable alternative, developing an asynchronous learning method is often complicated Stripelis et al. (2022); Tyagi & Swany (2023); Huba et al. (2022), set aside the lack of theoretical understanding in many cases.[1]

Based on the dilemma between orchestrating (or not) workers in a distributed system, this work focuses on a relatively simple problem: the Principal Component Analysis (PCA) Pearson (1901); Hotelling (1933); Wold et al. (1987); Majumdar (2009); Wang et al. (2013); d'Aspremont et al. (2007); Jiang et al. (2011); Zou et al. (2006) and its distributed implementation. Despite its "simplicity", the quest for a distributed PCA algorithm is still an active research area. It has recently gained momentum with the EigenGame implementation Gemp et al. (2020). EigenGame shows strides toward optimizing PCA for distributed computing environments, with ideas borrowed from game theory and implementations that mimic centralized versions. To retain its theoretical guarantees, EigenGame follows a strict hierarchy where each worker is responsible for a single component, and all workers respect hierarchy by waiting for their "superior" principal components (i.e., eigencomponents associated with larger eigenvalues) to be adequately estimated.[2]

**Our approach and contributions.** This work advances distributed PCA by building upon a collaborative computation model as in Gemp et al. (2020). Unlike traditional distributed PCA approaches that mimic centralized algorithms, our method innovates by allowing parallel computation of eigenvectors without strict sequential dependencies among the workers. This paradigm shift not only addresses the inherent inefficiencies of previous methods but also enhances the scalability and convergence speed. Herein, we delineate our primary contributions:

- *Novel Algorithmic Framework*: We propose a novel distributed PCA algorithm that fundamentally changes the computational dynamics. Using the covariance matrix, our approach enables multiple workers to perform eigenvector calculations in parallel. This method diverges from the traditional sequential computation models, significantly reducing total computation time.

- *Extension to Stochastic PCA*: in cases where the covariance is unknown or cannot be efficiently estimated, our algorithm can be easily modified to accommodate data that comes in mini-batches.

- *Theoretical Advancements*: We provide a robust theoretical framework that validates the convergence properties of our proposed algorithm. By formalizing the interaction between parallel computations and convergence rates, we establish a new theoretical benchmark for distributed PCA algorithms. This contribution underscores our algorithm's efficiency and enhances the understanding of parallel deflation processes in PCA.

- *Empirical Validation*: Through extensive experiments, we demonstrate the practical efficacy of our algorithm. Our results show that our approach at least meets the performance of existing baseline algorithms even on datasets as large as ImageNet Deng et al. (2009). These experiments substantiate our theoretical claims and highlight the real-world applicability of our method.

These contributions mark a significant step forward in distributed computing for PCA, providing theoretical insights and practical tools for data analysis applications.

## 1.1 RELATED WORKS

**Centralized approaches.** Principal Component Analysis (PCA) has been a cornerstone of statistical data analysis since 1901 Pearson (1901). Hotelling later expanded on Pearson's work, formalizing PCA within a multivariate analysis framework Hotelling (1933). Classical PCA typically involves the eigendecomposition of the data covariance matrix Jolliffe (2002).

With the advent of large datasets, iterative and gradient-based methods for PCA have gained prominence. These methods are particularly advantageous for large-scale data, where traditional eigendecomposition becomes computationally impractical. Krasulina and Oja & Karhunen proposed two of the earliest stochastic gradient descent methods for online PCA Krasulina (1969); Oja & Karhunen (1985). The application of the least square minimization to the PCA has also received attention Miao & Hua (1998); Yang (1995); Bannour & Azimi-Sadjadi (1995); Kung et al. (1994). More recently, Arora et al. (2012) and Shamir (2015) have proposed efficient stochastic optimization meth-

---

[1]In fact, asynchrony has been a topic of debate in distributed neural network training, where asynchronous training often inherently suffers from lower accuracy compared to synchronized analogs, resulting in the dominance of synchronized methods in neural network training Campos et al. (2017); Chen et al. (2017).

[2]We note here that even in this case, theory in Gemp et al. (2020) does not characterize how approximate estimates in eigencomponents higher in the "hierarchy" affect calculations in subsequent estimates.

ods that adapt to the streaming model of data (stochastic) and focused on the theoretical guarantees of gradient-based methods in such non-convex scenarios; see also Boutsidis et al. (2014); Garber et al. (2015); Shamir (2016); Kim & Klabjan (2020). Other approaches include manifold methods Demidovich et al. (2024); Chen et al. (2024); Wang et al. (2023b); Absil et al. (2008), Frank-Wolfe methods Beznosikov et al. (2023), Gauss-Newton methods Zhou et al. (2023), coordinate descent methods Lei et al. (2016), accelerated methods Xu et al. (2018), as well as variants of the PCA problem itself Journée et al. (2010); Yuan & Zhang (2013); Han & Liu (2014); Kim & Klabjan (2019); Kim et al. (2019). Nevertheless, these methods are primarily designed as centralized algorithms.

**Distributed approaches.** The line of work in Kannan et al. (2014); Liang et al. (2014); Boutsidis et al. (2016); Fan et al. (2019) utilizes randomized linear algebra and singular value decompositions of randomized projections of data in a distributed setting, leading to favorable theoretical results. For the case of distributed multiple eigenvector/subspace computation, Li et al. (2021) consider the distributed truncated singular value decomposition (SVD) problem and rely on FedAvg ideas McMahan et al. (2017) with local iterations. There, each worker utilizes an Orthogonal Procrustes Transformation Schönemann (1966); Cape (2020) to estimate the multiple subspace problem. *However, this line of work assumes that the covariance matrix is known or can be efficiently estimated.*

For distributed *leading* principal component computation, Garber et al. (2017) consider the stochastic setting and replace the Power Iteration scheme with convex optimization motions for better efficiency. Huang & Pan (2020) proposes a round-efficient solution by leveraging the connection to Riemannian optimization; similarly, see Alimisis et al. (2021). Recently, Wang et al. (2023b) proposed a Riemannian gradient-type method that admits low per-iteration computational and communication costs and can be readily implemented in an asynchronous setting. Beyond the classical distributed setting, there are works on the Byzantine and adversarial scenario Charisopoulos & Damle (2022); Zari et al. (2022), the streaming case Allen-Zhu & Li (2017); Yu et al. (2017), shift-and-invert preconditioning approaches Garber et al. (2016), and coreset-based approaches Feldman et al. (2020).

The papers above consider the data-parallel setting, where the data is distributed across machines, and each worker solve for all the principal components with its local data. DeepMind's EigenGame Gemp et al. (2020) introduced a model-parallel approach, framing each principal component as a player in a collaborative game. EigenGame optimizes the utility of each vector sequentially using Riemannian gradient ascent, but is also extended to the distributed scenario, where players can maximize their utility simultaneously, resulting in a model-parallel algorithm where solving each principal component is distributed across machines. The paper provided a convergence proof for the sequential process where vectors are optimized in a hierarchical order. However, for the distributed version, they didn't analyze how approximate steps affect overall convergence. A later improvement Gemp et al. (2022) was proposed, but also lacks theoretical guarantees for the distributed setting.

Our work complements existing literature both theoretically and practically. Unlike sequential approaches, our method does not require the completion of previous principal component computations before proceeding to the next. Moreover, We provide a comprehensive convergence analysis, establishing a stronger theoretical foundation than EigenGame, while maintaining practical efficiency.

## 2 PROBLEM STATEMENT AND BACKGROUND

Let $\mathbf{Y} \in \mathbb{R}^{n \times d}$ be the matrix representing an aggregation of $n$ properly scaled, centered data points, each with $d$ features. The empirical covariance matrix is given by $\mathbf{\Sigma} = \mathbf{Y}^\top \mathbf{Y} \in \mathbb{R}^{d \times d}$. Let $\mathbf{u}_k^\star$ and $\lambda_k^\star$ be the $k$th eigenvector and eigenvalue of $\mathbf{\Sigma}$, with $\lambda_1^\star \geq \cdots \geq \lambda_d^\star$. Then $\mathbf{u}_k^\star$ is the $k$th principal component of the data matrix $\mathbf{Y}$. Therefore, when $\mathbf{\Sigma}$ can be easily computed, principal component analysis aims at finding the top-$K$ eigenvectors of the empirical covariance matrix $\mathbf{\Sigma}$, where $K \leq d$.

**The leading eigenvector problem.** Finding the leading eigenvector is the cornerstone of finding multiple eigenvectors, and is thus utilized by many PCA algorithms. Mathematically, the problem of finding the leading eigenvector $\mathbf{u}_1^\star$ can be formulated as the following optimization problem:

$$\mathbf{u}_1^\star = \arg\max_{\mathbf{v} \in \mathbb{R}^d : \|\mathbf{v}\|_2 = 1} \mathbf{v}^\top \mathbf{\Sigma} \mathbf{v}. \tag{1}$$

In practice, algorithms like power iteration and Hebb's rule are used to solve the leading eigenvector.

**Definition 1** (Power Iteration). *The power iteration algorithm* PowIter $(\mathbf{\Sigma}, \mathbf{v}, T)$ *outputs a vector* $\mathbf{x}_T$ *based on the following iterates:*

$$\hat{\mathbf{x}}_{t+1} = \mathbf{\Sigma} \mathbf{x}_t; \quad \mathbf{x}_{t+1} = \hat{\mathbf{x}}_{t+1} / \|\hat{\mathbf{x}}_{t+1}\|_2.$$

**Definition 2** (Hebb's Rule). *The Hebb's Rule* Hebb $(\boldsymbol{\Sigma}, \mathbf{v}, T)$ *with some fixed step size* $\eta$ *outputs a vector* $\mathbf{x}_T$ *based on the following iterates:*

$$\hat{\mathbf{x}}_{t+1} = \mathbf{x}_t + \eta \boldsymbol{\Sigma} \mathbf{x}_t; \quad \mathbf{x}_{t+1} = \hat{\mathbf{x}}_{t+1} / \|\hat{\mathbf{x}}_{t+1}\|_2.$$

Under mild assumptions, the output $\mathbf{x}_T$ of both the power iteration and Hebb's rule converges to the top eigenvector of the input matrix $\boldsymbol{\Sigma}$, as the number of steps $T \to \infty$. Notably, the power iteration enjoys a linear convergence rate Shamir (2015).

**Top-$K$ eigenvector using sequential deflation.** An extension of (1) is the top-$K$ eigenvector problem, where one aims to find $\mathbf{u}_1^\star, \dots \mathbf{u}_K^\star$. Since $\mathbf{u}_1^\star, \dots, \mathbf{u}_K^\star$ form an orthonormal set, finding the top-$K$ eigenvector can be mathematically formulated as:

$$\mathbf{U}^\star = [\mathbf{u}_1^\star, \dots, \mathbf{u}_K^\star] \in \underset{\mathbf{V} \in \{\mathbf{Q}_{:,:K} : \mathbf{Q} \in \mathrm{SO}(d)\}}{\arg\max} \langle \boldsymbol{\Sigma} \mathbf{V}, \mathbf{V} \rangle, \tag{2}$$

where $\mathrm{SO}(d)$ denotes the group of rotations about a fixed point in $d$-dimensional Euclidean space.

A classical way to solve (2) is through *deflation* Hotelling (1933). Deflation operates in the following manner. Once the top component $\mathbf{u}_1^\star$ is approximated, the matrix $\boldsymbol{\Sigma}$ undergoes further processing to reside in the subspace orthogonal to the one spanned by the first eigenvector. This process is iterated by finding the leading eigenvector as in (1) on the deflated matrix, resulting in an approximation of the second component $\mathbf{u}_2^\star$, and so forth, as described below:

$$\boldsymbol{\Sigma}_1 = \boldsymbol{\Sigma}; \quad \mathbf{v}_k = \texttt{Top1}(\boldsymbol{\Sigma}_k, \hat{\mathbf{v}}_k, T);$$

$$\boldsymbol{\Sigma}_{k+1} = \boldsymbol{\Sigma}_k - \mathbf{v}_k \mathbf{v}_k^\top \boldsymbol{\Sigma}_k \mathbf{v}_k \mathbf{v}_k^\top, \quad (3)$$

where $\texttt{Top1}(\boldsymbol{\Sigma}_k, \hat{\mathbf{v}}_k, T)$ abstractly denotes any iterative algorithm initialized at $\hat{\mathbf{v}}_k$ and returns a normalized approximation of the top eigenvector of the deflated matrix $\boldsymbol{\Sigma}_k$ after $T$ iterations of execution. Consider the eigendecomposition $\boldsymbol{\Sigma} = \sum_{k'=1}^d \lambda_{k'}^\star \mathbf{u}_{k'}^\star \mathbf{u}_{k'}^{\star\top}$. When $T \to \infty$ and $\texttt{Top1}(\boldsymbol{\Sigma}_k, \hat{\mathbf{v}}_k, T)$ solves the top eigenvector of $\boldsymbol{\Sigma}$ exactly,

---

**Algorithm 1** Parallel Deflation

**Require:** $\boldsymbol{\Sigma} \in \mathbb{R}^{d \times d}$; # of eigenvectors (workers) $K$; sub-routine for top eigenvector $\texttt{PCA}(\cdot, \cdot, \cdot)$; # of iterations $T$; global communication rounds $L \geq K$.

**Ensure:** Approximate eigenvectors $\{\mathbf{v}_k\}_{k=1}^K$.
1: **for** $k = 1, \dots, K$ **do**
2:      Randomly initialize $\hat{\mathbf{v}}_{k,\texttt{init}}$ with unit norm;
3: **end for**
4: **for** $\ell = 1, \dots, L$ **do**
5:      **parfor** $k = 1, \dots, K$ **do**
6:          **if** $k \leq \ell$ **then**
7:              Receive $\mathbf{v}_{1,\ell-1}, \dots, \mathbf{v}_{k-1,\ell-1}$
8:              $\boldsymbol{\Delta}_{k',\ell} = \mathbf{v}_{k',\ell-1} \mathbf{v}_{k',\ell-1}^\top \boldsymbol{\Sigma} \mathbf{v}_{k',\ell-1} \mathbf{v}_{k',\ell-1}^\top$
9:              $\boldsymbol{\Sigma}_{k,\ell} = \boldsymbol{\Sigma} - \sum_{k'=1}^{k-1} \boldsymbol{\Delta}_{k',\ell}$
10:             $\mathbf{v}_{k,\ell} \leftarrow \texttt{Top1}(\boldsymbol{\Sigma}_{k,\ell}, \mathbf{v}_{k,\ell-1}, T)$
11:             Broadcast $\mathbf{v}_{k,\ell}$
12:          **else**
13:              $\mathbf{v}_{k,\ell} := \hat{\mathbf{v}}_{k,\texttt{init}}$;
14:          **end if**
15:      **end parfor**
16: **end for**
17: **return** $\{\mathbf{v}_{k,L}\}_{k=1}^K$

---

one can show that $\boldsymbol{\Sigma}_k = \sum_{k'=k}^d \lambda_{k'}^\star \mathbf{u}_{k'}^\star \mathbf{u}_{k'}^{\star\top}$ and $\mathbf{v}_k = \mathbf{u}_k^\star$. However, when $T$ is finite, it is shown in Liao et al. (2023) that each $\texttt{Top1}(\boldsymbol{\Sigma}_k, \hat{\mathbf{v}}_k, T)$ produces a non-negligible error that accumulates and propagates through the deflation process.

**Stochastic algorithm to find top-1 principal component.** When the dataset becomes large, the covariance matrix $\boldsymbol{\Sigma}$ may not be efficiently computed, making the previous routine of first computing the covariance matrix and then its eigenvector infeasible. Alternatively, people estimate $\boldsymbol{\Sigma}$ with $\hat{\boldsymbol{\Sigma}} = \hat{\mathbf{Y}}^\top \hat{\mathbf{Y}}$, where $\hat{\mathbf{Y}}$ is a mini-batch of the dataset. In this case, Hebb's rule can be written as

$$\hat{\mathbf{x}}_{t+1} = \mathbf{x}_t + \eta \hat{\mathbf{Y}}^\top (\hat{\mathbf{Y}} \mathbf{x}_t); \mathbf{x}_{t+1} = \hat{\mathbf{x}_{t+1}} / \|\hat{\mathbf{x}_{t+1}}\|_2$$

Notice that the stochastic estimate of the covariance matrix $\hat{\boldsymbol{\Sigma}}$ is never explicitly computed.

## 3 PARALLEL DEFLATION ALGORITHM

Here we introduce our algorithm that computes the top-$K$ principal components in a distributed environment. Our algorithm allows for parallel computation, significantly accelerating the process by overcoming the inherent sequential dependencies of traditional deflation techniques.

**Algorithm overview.** In our framework, we assign the task of computing each of the $K$ principal components to $K$ distinct workers. Each worker $k$ is responsible for computing the $k$th

principal component. The crux of our method lies in the modification of the traditional deflation process to suit a distributed setting. In a typical sequential deflation, the matrix required for computing the $k$th eigenvector is derived *only after obtaining the first $k-1$ eigenvectors*. This sequential dependency restricts parallelism. Our approach introduces an innovative twist:

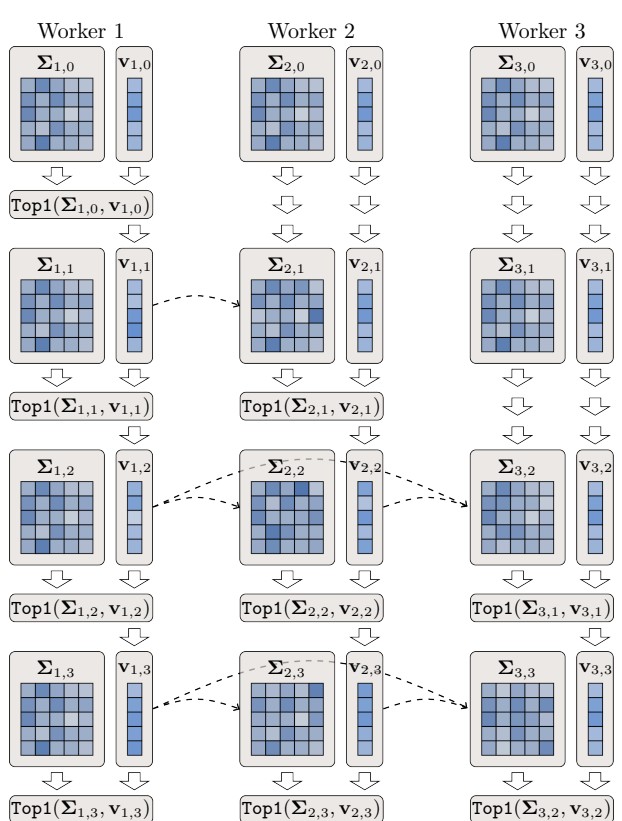

Figure 1: Illustration of the parallel deflation algorithm.

–*Initial Estimation*: Each worker $k$ begins by computing an inexact version of the $k$th deflated matrix, using preliminary estimates of the first $k-1$ eigenvectors produced by the corresponding workers.

–*Iterative Refinement*: Concurrently, workers refining the first $k-1$ eigenvectors provide updated estimates to worker $k$. This enables worker $k$ to refine the deflated matrix iteratively and improve the accuracy of the $k$th eigenvector estimation.

The main idea of introducing parallelism into the computation scheme is that worker $k$ does not wait for the first $k-1$ eigenvectors to be fully solved before solving the $k$th eigenvector. The parallel deflation algorithm is given in Algorithm 1.

The whole computation process is divided into $L$ communication rounds (**Line 4**). In the $\ell$th communication round, the $k$th worker will compute an approximation of the $k$th principal component $\mathbf{v}_{k,\ell}$ by running their own sub-routine in parallel, following the rule that the $k$th worker only deflates its matrix and start computing the $k$th principal component after the first $k-1$ workers have computed some rough estimation of the first $k-1$ principal components (**Lines 7-10**). Therefore, in the $\ell$th communication round, there can be two scenarios for worker $k$: $i$) if $\ell < k$, this means that not all of the first $k-1$ workers have computed some approximation of their own principal component. Therefore, worker $k$ does not deflate the matrix and output $\mathbf{v}_{k,\ell} = \hat{\mathbf{v}}_{k,\texttt{init}}$; $ii$) If $\ell \geq k$, then the first $k-1$ workers have at least computed one approximation of their own principal component. In this case, worker $k$ deflates the matrix using the most updated vectors $\mathbf{v}_{1,\ell-1}, \dots \mathbf{v}_{k-1,\ell-1}$ (**Line 7**), compute its approximation of the $k$th principal component by calling the $\texttt{Top1}(\cdot)$ on the deflated matrix starting from its output in the previous communication round (**Line 10**), and then broadcast the current approximation to the other workers for the next communication round (**Line 11**). An illustration of the algorithm is given in Figure 1.

**Extension to Stochastic PCA.** The algorithm described above can be applied to the case where the covariance matrix is either known or can be efficiently estimated. However, in many machine learning scenarios, the covariance matrix may not be direcly accessible. For instance, when data drawn from an underlying distribution comes in a streaming fashion Allen-Zhu & Li (2017), the traditional approach of first estimating the covariance matrix and then solves for its eigenvector is no longer efficient. Moreover, for large datasets that contains hundreds of thousands of features, it is impossible to compute or even store the covariance matrix Gemp et al. (2022; 2020). In these cases, our algorithm can be adapted to estimate the principal components in a stochastic fashion.

Let $\hat{Y}$ denote the mini-batch that the algorithm receives in the $t$th iteration. Starting from **Line 8**, whose major computation burden is on $\mathbf{v}_{k',\ell-1}^\top \mathbf{\Sigma} \mathbf{v}_{k',\ell-1}$, we notice that the covariance matrix is

estimated as $\mathbf{\Sigma} \approx \hat{\mathbf{\Sigma}} = \hat{\mathbf{Y}}^\top \hat{\mathbf{Y}}$. In this case, we have:

$$\mathbf{v}_{k',\ell-1}^\top \hat{\mathbf{\Sigma}} \mathbf{v}_{k',\ell-1} = \mathbf{v}_{k',\ell-1}^\top \hat{\mathbf{Y}}^\top \hat{\mathbf{Y}} \mathbf{v}_{k',\ell-1} = \|\hat{\mathbf{Y}} \mathbf{v}_{k',\ell-1}\|_2^2.$$

Therefore, each $\mathbf{\Delta}_{k',\ell}$ in Algorithm 1 can be written as $\mathbf{\Delta}_{k',\ell} = \|\hat{\mathbf{Y}} \mathbf{v}_{k',\ell-1}\|_2^2 \mathbf{v}_{k',\ell-1} \mathbf{v}_{k',\ell-1}^\top$. Thus, **Line 9** becomes:

$$\mathbf{\Sigma}_{k,\ell} \approx \hat{\mathbf{\Sigma}}_k = \hat{\mathbf{Y}}^\top \hat{\mathbf{Y}} - \sum_{k'=1}^{k-1} \|\hat{\mathbf{Y}} \mathbf{v}_{k',\ell-1}\|_2^2 \mathbf{v}_{k',\ell-1} \mathbf{v}_{k',\ell-1}^\top.$$

This new form of $\mathbf{\Sigma}_{k,\ell}$ allows an efficient estimation of the matrix-vector product $\mathbf{\Sigma}_{k,\ell} \mathbf{x}$, as in:

$$\hat{\lambda}_{k'} = \|\hat{\mathbf{Y}} \mathbf{v}_{k'}\|_2^2; \quad \forall k' \in [k-1]; \quad \mathbf{\Sigma}_{k,\ell} \mathbf{x} = \hat{\mathbf{Y}}^\top \hat{\mathbf{Y}} \mathbf{x}_t - \sum_{k'=1}^{k-1} \hat{\lambda}_{k'} \left( \mathbf{v}_{k',\ell-1}^\top \mathbf{x}_t \right) \cdot \mathbf{v}_{k',\ell-1}. \quad (4)$$

In the current form of Algorithm 1, the computation of **Lines 8-9** takes $O\left(Kd^2\right)$ time. Moreover, calling the $\mathrm{Top1}$ function in **Lines 10**, any matrix-vector multiplication $\mathbf{\Sigma}_{k,\ell} \mathbf{x}$ will take $O\left(d^2\right)$ time. Notice that in (4), the complexity of computing each $\mathbf{Y} \mathbf{v}_{k'}$ is $O\left(nd\right)$. Thus computing $\hat{\lambda}_{k'}$ takes $O\left(nd\right)$. In total,(4) has a complexity of $O\left(Knd\right)$. In (4), computing the first term $\hat{\mathbf{Y}}^\top \hat{\mathbf{Y}} \mathbf{x}$ involves computing first $\mathbf{y}_t = \hat{\mathbf{Y}} \mathbf{x}_t$, which takes $O\left(nd\right)$, and then $\hat{\mathbf{Y}}^\top \mathbf{y}_t$, which also takes $O\left(nd\right)$. Thus, computing the first term $\hat{\mathbf{Y}}^\top \hat{\mathbf{Y}} \mathbf{x}_t$ takes $O\left(nd\right)$ in total. For the second term, each summand takes $O(d)$ to compute, giving the complexity of computing the second term as $O\left(kd\right)$. Therefore, each iteration of (4) takes $O\left((n+k)d\right)$. This implies a saving in the computation cost, since in this case, $n$ is the batch size and can be much smaller than $d$. The complete algorithm in the stochastic setting is given in Algorithm 2 in the Appendix.

**Connection with EigenGame.** The EigenGame Gemp et al. (2020) considers the problem of solving the top-$K$ eigenvectors of a matrix as a game between $K$ players, with the $k$th player solving $\mathbf{v}_k$ by maximizing its utility: $\mathbf{v}_k = \arg\max_{\mathbf{v}:\|\mathbf{v}\|_2=1} \mathcal{U}_k \left( \mathbf{v} \mid \mathbf{v}_1, \ldots \mathbf{v}_{k-1} \right)$, where:

$$\mathcal{U}_k \left( \mathbf{v} \mid \{\mathbf{v}_{k'}\}_{k'=1}^{k-1} \right) = \mathbf{v}^\top \mathbf{\Sigma} \mathbf{v} - \sum_{k'=1}^{k-1} \frac{\left( \mathbf{v}_{k'}^\top \mathbf{\Sigma} \mathbf{v} \right)^2}{\mathbf{v}_{k'}^\top \mathbf{\Sigma} \mathbf{v}_{k'}}. \quad (5)$$

Similarly, the deflation algorithm in (2) also bears a game formulation, where the utility of the $k$th player is given by:

$$\mathcal{V}_k \left( \mathbf{v} \mid \{\mathbf{v}_{k'}\}_{k'=1}^{k-1} \right) = \mathbf{v}^\top \left( \mathbf{\Sigma} - \sum_{k'=1}^{k-1} \mathbf{v}_{k'} \mathbf{v}_{k'}^\top \mathbf{\Sigma} \mathbf{v}_{k'} \mathbf{v}_{k'}^\top \right) \mathbf{v} = \mathbf{v}^\top \mathbf{\Sigma} \mathbf{v} - \sum_{k'=1}^{k-1} \mathbf{v}_{k'}^\top \mathbf{\Sigma} \mathbf{v}_{k'} \cdot \left( \mathbf{v}_{k'}^\top \mathbf{v} \right)^2. \quad (6)$$

It should be noted that both the EigenGame utility $\mathcal{U}_k$ and the deflation utility $\mathcal{V}_k$ depend on only the policy of the first $k-1$ players. Moreover, when the first $k-1$ players recovers the top-$(k-1)$ eigenvectors exactly, we shall have:

$$\mathcal{V}_k \left( \mathbf{v} \mid \{\mathbf{u}_{k'}^\star\}_{k'=1}^{k-1} \right) = \mathbf{v}^\top \mathbf{\Sigma} \mathbf{v} - \sum_{k'=1}^{k-1} \lambda_{k'}^\star \left( \mathbf{v}^\top \mathbf{u}_{k'}^\star \right)^2 = \mathcal{U}_k \left( \mathbf{v} \mid \{\mathbf{u}_{k'}^\star\}_{k'=1}^{k-1} \right).$$

To this end, we can also show that the set of true eigenvectors $\{\mathbf{u}_k^\star\}_{k=1}^K$ is the unique strict Nash Equilibrium defined by the utilities in (6). The proof of Theorem 1 is deferred to Appendix A.

**Theorem 1.** *Assume that the covariance matrix $\mathbf{\Sigma}$ has positive and strictly decreasing eigenvalues $\lambda_1^\star > \cdots > \lambda_K^\star > 0$. Then, $\{\mathbf{u}_k^\star\}_{k=1}^K$ is the unique strict Nash Equilibrium defined by the utilities in (6) up to sign perturbation, i.e., replacing $\mathbf{u}_k^\star$ with $-\mathbf{u}_k^\star$.*

## 4 CONVERGENCE GUARANTEE FOR PARALLEL DEFLATION ALGORITHM

We provide a convergence guarantee for the parallel deflation algorithm in Algorithm 1. The pivot of the convergence analysis will be to track the dynamics of $\{\mathbf{\Sigma}_{k,\ell}\}_{k=1}^K$ and $\{\mathbf{v}_{k,\ell}\}_{k=1}^K$ as $\ell$ increases. The dynamics of the two sequences from Algorithm 1 can be compactly represented as:

$$\mathbf{\Sigma}_{k,\ell} = \mathbf{\Sigma} - \sum_{k'=1}^{k-1} \mathbf{v}_{k',\ell-1} \mathbf{v}_{k',\ell-1}^\top \mathbf{\Sigma} \mathbf{v}_{k',\ell-1} \mathbf{v}_{k',\ell-1}^\top; \quad \mathbf{v}_{k,\ell} = \mathrm{Top1} \left( \mathbf{\Sigma}_{k,\ell}, \mathbf{v}_{k,\ell-1} \right); \quad \forall \ell \geq k.$$

Here, we embed the number of solver steps $T$ in the property of the abstract local solver $\mathrm{Top1}(\cdot)$. Indeed, if $\mathrm{Top1}(\cdot)$ returns the exact top eigenvector of the input matrix every time it is called, then

we can easily see that $\mathbf{v}_{k,\ell} = \mathbf{u}_k^\star$ for all $\ell \geq k$. When $\mathtt{Top1}(\cdot)$ returns an inexact estimate of the input matrix sequentially, i.e., worker $k$ waits until the top-$(k-1)$ worker no longer improves the estimation of the top-$(k-1)$ eigenvectors, the error is analyzed by Liao et al. (2023).

Our scenario is further complicated by the continuous improvement of the eigenvector estimates used to deflate the matrix: in each communication round, the $\mathtt{Top1}(\cdot)$ function, called by worker $k$, will start at the estimate of the top eigenvector of the deflated matrix in the previous round but will be fitted to the top eigenvector of the deflated matrix in the current round. Our convergence analysis tackles this complicated dynamic by utilizing the following abstraction of the $\mathtt{Top1}(\cdot)$ sub-routine.

**Assumption 1.** *Let $\hat{\boldsymbol{\Sigma}} \in \mathbb{R}^{d \times d}$ be a real symmetric matrix. Let $\lambda^\star$ be its eigenvalue with the largest absolute value, and let $\mathbf{u}^\star$ be the corresponding eigenvector of $\lambda^\star$. We assume that there exists a real value $\mathcal{F}\left(\hat{\boldsymbol{\Sigma}}\right) \in (0,1)$ that depends on $\hat{\boldsymbol{\Sigma}}$ such that for any $\mathbf{x}_0 \in \mathbb{R}^d$, $\mathtt{Top1}(\cdot)$ satisfies:*

$$\left\|\mathtt{Top1}\left(\hat{\boldsymbol{\Sigma}}, \mathbf{x}_0\right) - \mathbf{u}^\star\right\|_2 \leq \mathcal{F}\left(\hat{\boldsymbol{\Sigma}}\right)\|\mathbf{x}_0 - \mathbf{u}^\star\|_2.$$

Assumption 1 can be easily guaranteed. as long as the $\mathtt{Top1}(\cdot)$ algorithm enjoys a non-asymptotic convergence to the top eigenvector; see the Related Works section above. With Assumption 1, the convergence of Algorithm 1 is given by the following theorem.

**Theorem 2.** *Assume that Assumption 1 holds, and let $\mathcal{F}_k = \max_{\ell \geq k} \mathcal{F}\left(\hat{\Sigma}_{k,\ell}\right)$. Let $W_{-1}(\cdot)$ be the Lambert-W function in the $-1$ branch[3], and define for $a > 0$:*

$$\hat{W}(a) = \begin{cases} -W_{-1}(-a) & \text{if } a \in (0, e^{-1}) \\ 1 & \text{if } a \in [e^{-1}, \infty) \end{cases}$$

*Let $\{m_k\}_{k=0}^K$ be a sequence of numbers denoting the convergence rates of recovering the $K$ eigenvectors, where $m_k = \max\left\{\mathcal{F}_k, \frac{1}{k} + \frac{k-1}{k}m_{k-1}\right\}$ and $m_0 = 1$ as a dummy starting point. Let $\{s_k\}_{k=1}^n$ be a sequence of integers denoting the starting communication round where the $K$ eigenvectors' error recovery enters the linear convergence phase, respectively. To be more specific, let $s_1 = 1$ and for all $k \in [K-1]$ and $k' \in [k]$:*

$$s_{k+1} \geq \max\left\{\frac{\hat{W}\left(m_k \log 1/m_k\right)}{\log 1/m_k}, \frac{km_k + 1}{1 - m_k}\right\} + \frac{\hat{W}\left(\frac{\lambda_{k+1}^\star - \lambda_{k+2}^\star}{12k\lambda_{k'}^\star}\left(\log 1/m_k\right)^2\right)}{\log 1/m_{k'}} + s_{k'}. \quad (7)$$

*Then, we have that the following holds for all $k \in [K]$*

$$\|\mathbf{v}_{k,\ell} - \mathbf{u}_k^\star\|_2 \leq 6\left(\ell - s_k + 2\right)m_k^{\ell - s_k + 1}; \quad \forall \ell \geq s_k - 1. \quad (8)$$

In words, Theorem 2 says that starting from the $s_k$th communication round, the recovery error of the $k$th eigenvector converges according to a nearly-linear convergence rate given in (8). However, the convergence starting point $s_k$ for the $k$th eigenvector must be later than the convergence starting point for the $1, \ldots, k-1$th eigenvector for a number of communication rounds. This delay in the convergence starting point is characterized in (7). Intuitively, the starting point $s_k$ denotes the index of the communication round where the top-$(k-1)$ eigenvectors have been estimated accurately enough for the $k$th worker to make positive progress.

**Remark 1.** By the definition that $m_k = \max\left\{\mathcal{F}_k, \frac{1}{k} + \frac{k-1}{k}m_{k-1}\right\}$, one could see that $m_k < 1$ since $\mathcal{F}_k < 1$ for all $k \in [K]$. The convergence rate in (8) involves the product of a linear term $\ell - s_k + 2$ and an exponential term $m_k^{\ell - s_k + 1}$. When $\ell$ is large enough, $m_k^{\ell - s_k + 1}$ decays at a much faster speed than the increase of $\ell - s_k + 2$, thus giving a nearly-linear convergence rate.

**Remark 2.** Upper bound on the separation between the $s_k$'s. By using the inequality that $W_{-1}(e^{-u-1}) \geq -1 - \sqrt{2u} - u$ Chatzigeorgiou (2013), we could obtain that $\hat{W}(a) \leq \log 1/a + \sqrt{2(\log 1/a - 1)} + 1$ when $a \in (0, e^{-1})$. Therefore, we can conclude that $\hat{W} = O(\max\{1, \log 1/a\})$. Notice that (7) requires that $s_{k+1}$, the starting point of the linear convergence for the error of $\mathbf{v}_{k+1,\ell}$, must be later than $s_1, \ldots, s_k$ for some steps. Using the bound of $\hat{W}(a)$, one could simplify (7) to:

$$s_{k+1} \geq s_k + O\left(\max\left\{\left(\log \frac{1}{m_k}\right)^{-1}\left(1 + \log \frac{k\lambda_k^\star}{\lambda_{k+1}^\star - \lambda_{k+2}^\star}\right), \frac{km_k + 1}{1 - m_k}\right\}\right).$$

---

[3]The Lambert-W function in the $-1$-branch is defined as the inverse of the function $f(x) = xe^x$ when $x \in (-\infty, -1)$.

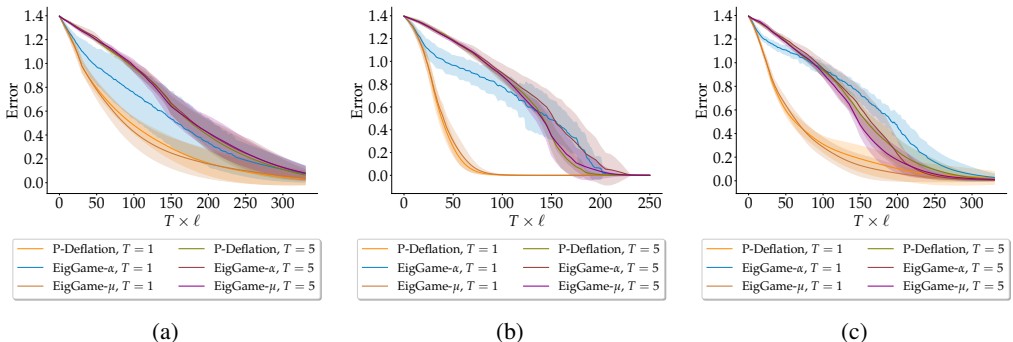

(a)            (b)            (c)

Figure 2: Comparison of the convergence behavior of parallel deflation, EigenGame-$\alpha$, and EigenGame-$\mu$ in deterministic setting on (a). synthetic dataset with power-law decaying eigenvalues, (b). synthetic dataset with exponentially decaying eigenvalues, and (c). MNIST dataset.

This simplification implies that a smaller $m_k$ would cause a smaller decay between $s_k$ and $s_{k+1}$. Since $m_k$ depends on $\mathcal{F}_k$, a smaller $m_k$ can be achieved by doing more local steps in the call to $\texttt{Top1}(\cdot)$. However, since the decay between $s_k$ and $s_{k+1}$ is measured in terms of the communication rounds, doing more local steps also increases the computation time per round in the delayed periods.

**Sketch of Proof.** The key challenge in proving Theorem 2 lies in handling the dynamic, asynchronous nature of our algorithm. Unlike sequential deflation, where each principal component is computed after the previous ones have converged and stays fixed, our method deals with simultaneous updates of all principal components. This requires careful analysis of how errors propagate and accumulate across different workers. To start, we derive upper bounds of the per-iteration difference between the actual deflated matrix $\mathbf{\Sigma}_k$ and the ideal deflated matrix $\mathbf{\Sigma}_k^*$ and the difference between the estimated eigenvector $\mathbf{v}_{k,\ell}$ and the ground-truth eigenvector $\mathbf{u}_k^*$. Notice that these two upper bounds are inter-dependent. We apply Davis-Kahan Sin$\Theta$ Theorem Davis & Kahan (1970) to derive the bound between the matrices' top-eigenvectors based on the matrix differences. Next, we carefully choose a convergence starting point $s_k$ for each eigenvector. Construct two simpler two-dimensional sequences $\{B_{k,\ell}\}$ and $\{G_{k,\ell}\}$ starting from $s_k$'s that upper bound these differences. Lastly, we unroll the bounds on $\{B_{k,\ell}\}$ and $\{G_{k,\ell}\}$ to arrive at a closed form upper bound on error of the estimated eigenvector $\mathbf{v}_{k,\ell}$. The detailed proof is deferred to Appendix B.

## 5 EXPERIMENTS

In this section, we experimentally verify the performance of the parallel deflation algorithm.

**Baseline algorithms.** We compared the parallel deflation algorithm with power iteration as the $\texttt{Top1}$ subroutine with the distributed version of EigenGame-$\alpha$ Gemp et al. (2020) and EigenGame-$\mu$ Gemp et al. (2022). It should be noticed that EigenGame-$\alpha$ was proposed as a sequential principal component recovery algorithm and can be adapted as a distributed algorithm. Moreover, both EigenGame-$\alpha$ and EigenGame-$\mu$ are restricted to the case of one iteration of update per communication round. We modified their algorithm to generalize to multiple iterations of update in Algorithm 3 and Algorithm 4 in Appendix D. As in the implementation of Gemp et al. (2020) ad Gemp et al. (2022), we do not project the utility gradient to the unit sphere.

**Evaluation Metric.** We evaluate the performance of the three algorithms by computing how close the recovered principal component is to the true eigenvector of the covariance matrix. In particular, notice that if a vector $\mathbf{u}_k^\star$ is the $k$th principal component (equivalently $k$th eigenvector of the covariance matrix), then $-\mathbf{u}_k^\star$ is also the $k$th principal component. Therefore, for the set of true principal components $\{\mathbf{u}_k^\star\}_{k=1}^K$ and a set of recovered principal component $\{\mathbf{v}_k\}_{k=1}^K$, we use the following metric to compute the approximation error

$$\mathcal{E}\left(\{\mathbf{u}_k^\star\}_{k=1}^K, \{\mathbf{v}_k\}_{k=1}^K\right) = \left(\frac{1}{K}\sum_{k=1}^K \min_{s \in \{\pm 1\}} \|\mathbf{u}_k^\star - s \cdot \mathbf{v}_k\|_2^2\right)^{\frac{1}{2}} \tag{9}$$

**Deterministic Experiments.** For synthetic experiments, we choose the number of features $d = 1000$, which gives the covariance matrices $\mathbf{\Sigma} \in \mathbb{R}^{1000 \times 1000}$. We consider $\mathbf{\Sigma}$ generated with two different eigenvalue spectra: $i)$ a power-law decaying spectrum $\lambda_k^\star = \frac{1}{\sqrt{k}}$, and $ii)$ an exponential

decaying spectrum $\lambda_k^\star = \frac{1}{1.1^k}$. We choose the number of local updates in each communication round $T$ to be $T \in \{1, 5\}$. We ran parallel deflation, EigenGame-$\alpha$, and EigenGame-$\mu$ to recover the top-30 eigenvectors ($K = 30$). For each setting, we run 10 trials with different random initialization.

Figure 2a presents the convergence behavior of the three algorithms with $T \in \{1, 5\}$ on the synthetic matrix with $\lambda_k^\star = \frac{1}{\sqrt{k}}$. Both EigenGame-$\mu$ and parallel deflation demonstrate stable convergence to a low error value under the case of $T = 1$ and $T = 5$, with parallel deflation converging slightly slower than EigenGame-$\mu$ in the first 200 total steps, and then arriving at a lower error than EigenGame-$\mu$ in the last 100 total steps. On the other hand, EigenGame-$\alpha$ shows unstable convergence that appears to be significantly slower than EigenGame-$\mu$ and parallel deflation. Figure 2b presents the result on the synthetic matrix with $\lambda_k^\star = \frac{1}{1.1^k}$. In general, the convergence behavior is similar to Figure 2a, with parallel deflation and EigenGame-$\mu$ exhibiting a fast and stable convergence, and EigenGame-$\alpha$ being slower. In both Figure 2a and Figure 2b, the setting of $T = 1$ shows a faster convergence than $T = 5$. This is because $T = 1$ allows more communication in a fixed number of total steps $T \times L$, which keeps the deflated matrices of each local worker to be better updated.

We also use the real-world dataset of MNIST, which contains $n = 60000$ hand-written digits, to compute the covariance. In particular, each $28 \times 28$ image is first divided by 255 for numerical stability and then flattened into a vector $\mathbf{y}_i$ of $d = 784$ features. Similar to the synthetic experiments, we also choose $T \in \{1, 5\}$ and aim at recovering the top-30 eigenvectors. The result on the MNIST dataset is presented in Figure 2c. Similar convergence behavior of the three algorithms, where parallel deflation and EigenGame-$\mu$ demonstrate similar convergence speed, and EigenGame-$\alpha$ converges much slower. Noticeably, for EigenGame-$\alpha$, we could observe that the case $T = 1$ converges even slower than the case $T = 5$. We hypothesize that this is because one local iteration is not sufficient for the top eigenvector solvers to provide an accurate enough estimate for the following solvers to make positive progress.

**Stochastic Setting.** In the stochastic experiments, we first test the algorithm's performance on a synthetic Gaussian distribution. We generate a covariance matrix $\mathbf{\Sigma}$ with power-law decaying spectrum as in the deterministic experiments. However, instead of directly passing it to the PCA solver, we sample I.I.D. samples from $\mathcal{N}(\mathbf{0}, \mathbf{\Sigma})$ and pass the sampled data batches to parallel deflation and EigenGame in a streaming fashion. We use a decaying step size for all three algorithms, and the result is given in Figure 3a. In this setting, parallel deflation shows a slightly worse performance than the two EigenGame algorithms. We hypothesis that this is because parallel deflation is more sensitive to the step size tuning in the stochastic case.

In Figure 3b, we plot the performance of parallel deflation and EigenGame in the stochastic setting of the MNIST dataset. That is, batches of the MNIST training set is sampled in each iteration and passed to the algorithms. We observe that parallel deflation achieves a similar performance to EigenGame-$\mu$, with a slightly faster convergence speed in the early phase of the algorithm.

In addition, we also test the performance of parallel deflation on the ImageNet dataset Deng et al. (2009) that contains 1.2M images. In particular, each $224 \times 224$ image is flattened into a vector of dimension 50176. The large scale of the dataset makes it impossible to even compute or store the covariance matrix on a single device. We use both parallel deflation and EigenGame-$\mu$ to compute the top-10 eigenvector of the dataset. It should be noted that since no "ground-truth" principal component is know, we can no longer use the metric in (9). Instead, we notice that, since finding the $k$th eigenvector can be seen as maximizing $\mathbf{v}^\top \mathbf{\Sigma} \mathbf{v}$ given that $\mathbf{v}$ is orthogonal to all previous eigenvectors, we can use an aggregation of the terms $\mathbf{v}_k^\top \mathbf{\Sigma} \mathbf{v}_k$ as a metric to evaluate the quanlity of the solved principal components. To follow the internal hierarchy of the eigenvectors that the leading eigenvectors are free to explore more space and thus are expected to attain a larger $\mathbf{v}^\top \mathbf{\Sigma} \mathbf{v}$, we penalize terms with larger index with a discounting factor. This result in the following metric

$$\mathcal{M}\left(\{\mathbf{v}_k\}_{k=1}^K\right) = \sum_{k=1}^K \frac{1}{k} \mathbf{v}_k^\top \mathbf{\Sigma} \mathbf{v}_k = \frac{1}{n} \sum_{i=1}^n \sum_{k=1}^K \frac{\left(\bar{\mathbf{y}}_i^\top \mathbf{v}_k\right)^2}{k} \tag{10}$$

Notice that a larger value of $\mathcal{M}(\cdot)$ implies a better quality of the recovered eigenvectors. In practice, we compute this metric by using a batched aggregation of the terms $\frac{\left(\bar{\mathbf{y}}_i^\top \mathbf{v}_k\right)^2}{k}$ over all $i \in [n]$. We plot the result in Figure 3c, and we could observe that in this large scale dataset, our algorithm can keep up with the performance of the state-of-the-art algorithm EigenGame-$\mu$.

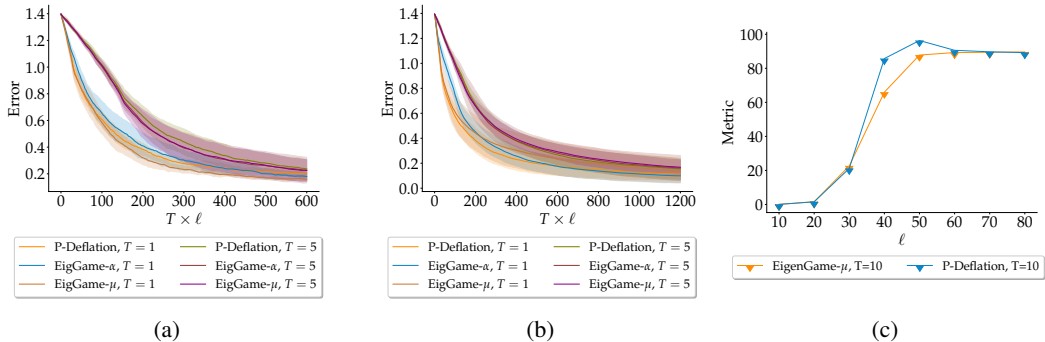

(a)                                    (b)                                    (c)

Figure 3: Comparison of the convergence behavior of parallel deflation, EigenGame-$\alpha$, and EigenGame-$\mu$ in stochastic setting on (a). synthetic dataset with power-law decaying eigenvalues, (b). MNIST dataset, and (c) ImageNet dataset.

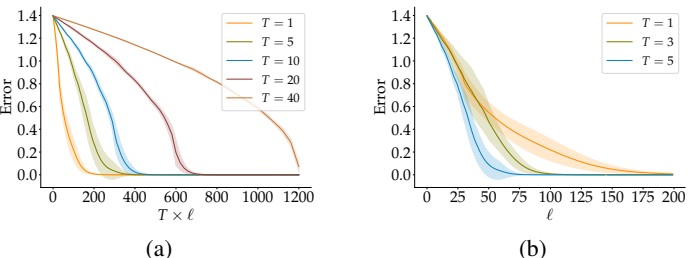

(a)                                    (b)

Figure 4: Ablation study of the parallel deflation algorithm. (a) shows the benefit of the run-time by increasing the parallelism. (b) shows the benefit of decreasing the communication cost by increasing the number of local iterations.

**Ablation studies.** We conducted additional ablation studies for the parallel deflation algorithms, with the results presented in Figure 4. In Figure 4a, we conduct additional experiments comparing how different choices of the number of local updates $T$ contribute to the convergence speed. In particular, since in each communication round, the local updates of all workers are done in parallel, $T \times \ell$ would represent the total time elapsed under the ideal scenario of no communication cost. We could see from Figure 4a that a smaller $T$ results in a faster convergence speed. Remarkably, since we choose $T \times L = 1200$ and aim at recovering 30 eigenvectors, the case where $T = 40$ corresponds to $L = 30$, demonstrating the convergence behavior of the sequential deflation algorithm. Figure 4a thus supports that introducing additional parallelism into the deflation algorithm indeed speeds up the computation process. On the other hand, Figure 4b considers the case where the communication cost is the major burden. In this case, we can run the parallel deflation algorithm with a larger number of local updates, hoping to make more progress within one communication round. Indeed, Figure 4b shows that a large number of local updates result in a faster convergence within a fixed number of communication rounds on larger datasets.

## 6    CONCLUSION

In this paper, we present a novel algorithmic framework for computing the principal components in a distributed fashion. Based on the classical deflation method to solve for the top-$K$ eigenvectors of a matrix, our algorithm distributes the workload of computing each eigenvector to a single worker. We introduce additional parallelism by early-starting the computation of the following eigenvectors based on the initial rough estimation of leading principal components and continuously refining the local deflated matrix based on updated estimated principal components. We show that our algorithmic framework has a similar game-theoretic formulation as the EigenGame, while enjoying a nice convergence guarantee even in the distributed case. Moreover, through experiments, we show that our algorithm converges at a comparable speed to the EigenGame-$\mu$, and is faster than the EigenGame-$\alpha$. Future work can focus on empirically examining the potential of using other `Top1` subroutines in our parallel deflation algorithm, such as Oja's rule.

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

## A    MISSING PROOF FROM SECTION 3

*Proof of Theorem 1.* Let $\mathbf{u}_1^\star, \ldots, \mathbf{u}_d^\star$ be the set of eigenvectors of $\boldsymbol{\Sigma}$, and $\lambda_1^\star, \ldots, \lambda_d^\star$ be the corresponding eigenvalues, potentially with some $\lambda_k = 0$. Recall that in the game formulation of the deflation algorithm, the utility function of the $k$th player is given by

$$
\mathcal{V}_k \left( \mathbf{v} \mid \{\mathbf{v}_{k'}\}_{k'=1}^{k-1} \right) = \mathbf{v}^\top \boldsymbol{\Sigma} \mathbf{v} - \sum_{k'=1}^{k-1} \mathbf{v}_{k'}^\top \boldsymbol{\Sigma} \mathbf{v}_{k'} \cdot \left( \mathbf{v}_{k'}^\top \mathbf{v} \right)^2
$$

and when $\mathbf{v}_{k'} = \mathbf{u}_{k'}^\star$ for $k' \in [k-1]$, we have

$$
\mathcal{V}_k \left( \mathbf{v} \mid \{\mathbf{u}_{k'}^\star\}_{k'=1}^{k-1} \right) = \mathbf{v}^\top \boldsymbol{\Sigma} \mathbf{v} - \sum_{k'=1}^{k-1} \lambda_{k'}^\star \cdot \left( \mathbf{v}_{k'}^\top \mathbf{v} \right)^2
$$

It should be noted that $\boldsymbol{\Sigma}$ has the eigendecomposition $\boldsymbol{\Sigma} = \sum_{k'=1}^d \lambda_{k'}^\star \mathbf{u}_{k'}^\star \mathbf{u}_{k'}^{\star\top}$. Therefore we can rewrite $\mathbf{v}^\top \boldsymbol{\Sigma} \mathbf{v}$ as $\sum_{k'=1}^d \lambda_{k'}^\star \left( \mathbf{v}^\top \mathbf{u}_{k'}^\star \right)^2$. Thus, the utility becomes

$$
\mathcal{V}_k \left( \mathbf{v} \mid \{\mathbf{u}_{k'}^\star\}_{k'=1}^d \right) = \sum_{k'=k}^{k-1} \lambda_{k'}^\star \left( \mathbf{v}^\top \mathbf{u}_{k'}^\star \right)^2
$$

Since $\mathbf{u}_1^\star, \ldots, \mathbf{u}_d^\star$ spans $\mathbb{R}^d$ and are mutually orthogonal, we can write $\mathbf{v} = \sum_{j=1}^d \beta_j \mathbf{u}_j^\star$. where $\sum_{j=1}^d \beta_j^2 = 1$. Then we have

$$
\mathcal{V}_k \left( \mathbf{v} \mid \{\mathbf{u}_{k'}^\star\}_{k'=1}^d \right) = \sum_{k'=k}^{k-1} \lambda_{k'}^\star \left( \sum_{j=1}^d \beta_j \mathbf{u}_j^{\star\top} \mathbf{u}_{k'}^\star \right)^2 = \sum_{k'=k}^{k-1} \lambda_{k'}^\star \beta_{k'}^2
$$

Since $\lambda_k$'s are strictly decreasing and positive, we must have that the maximum of $\mathcal{V}_k \left( \mathbf{v} \mid \{\mathbf{u}_{k'}^\star\}_{k'=1}^d \right)$ is only attained when $\beta_k^2 = 1$, which implies that $\mathbf{v} = \pm \mathbf{u}_k^\star$ will be the only optimal policies for player $k$.

The uniqueness can be shown by induction. To start, we notice that $\mathcal{V}_1$ does not depend on the policy of the other plays. Therefore, the only optimal policy for player 1 is $\mathbf{v}_1 = \pm \mathbf{u}_1^\star$. This shows the base case. Now, assume that within the top-$(k-1)$ players, the optimal policies are $\mathbf{v}_{k'} = \pm \mathbf{u}_{k'}^\star$ for $k' \in [k-1]$. By the formulation of the utility functions, these optimal policies are not affected by the policy of player $k, \ldots, K$. Moreover, it should be noted that the utility of player $k$ only depends on the top-$(k-1)$ players. Therefore, the optimal policy for player $k$ must be $\mathbf{v}_k = \pm \mathbf{u}_k^\star$. This finishes the inductive step and completes the proof. $\square$

## B    MISSING PROOF FROM SECTION 4

We first introduce a tool that we will utilize in the proof in this section.

**Lemma 1** (sin $\Theta$ Theorem Davis & Kahan (1970)). *Let $\mathbf{M}^* \in \mathbb{R}^{d \times d}$ and let $\mathbf{M} = \mathbf{M}^* + \mathbf{H}$. Let $\mathbf{a}_1^*$ and $\mathbf{a}_1$ be the top eigenvectors of $\mathbf{M}^*$ and $\mathbf{M}$, respectively. Then we have:*

$$
\sin \angle \{\mathbf{a}_1^*, \mathbf{a}_1\} \leq \frac{\|\mathbf{H}\|_2}{\min_{j \neq k} |\sigma_k^* - \sigma_j|}.
$$

### B.1    PROOF OF THEOREM 2

Define $\boldsymbol{\Sigma}_k^\star = \sum_{k=1}^d \lambda_k^\star \mathbf{u}_k^\star \mathbf{u}_k^{\top\star}$ as the "ground-truth" deflation matrix. Recall that the parallel deflation algorithm executes

$$
\boldsymbol{\Sigma}_{k,\ell} = \boldsymbol{\Sigma} - \sum_{k'=1}^{k-1} \mathbf{v}_{k',\ell-1} \mathbf{v}_{k',\ell-1}^\top \boldsymbol{\Sigma} \mathbf{v}_{k',\ell-1} \mathbf{v}_{k',\ell-1}^\top; \quad \mathbf{v}_{k,\ell} = \mathtt{Top1} \left( \boldsymbol{\Sigma}_{k,\ell}, \mathbf{v}_{k,\ell-1} \right) \quad (11)
$$

Let $\mathbf{u}_{k,\ell}$ denote the top eigenvector of $\mathbf{\Sigma}_{k,\ell}$. In particular, it suffices to show that the quantity $\|\mathbf{v}_{k,\ell} - \mathbf{u}_k^\star\|_2^2$ decreases as $\ell$ increases. Combining Assumption 1 and the definition that $\mathcal{F}_k = \max_{\ell \geq k} \mathcal{F}(\mathbf{\Sigma}_{k,\ell})$, we have that

$$\|\mathbf{v}_{k,\ell} - \mathbf{u}_{k,\ell}\|_2 \leq \mathcal{F}_k \|\mathbf{v}_{k,\ell-1} - \mathbf{u}_{k,\ell}\|_2 \tag{12}$$

We could upper bound $\|\mathbf{v}_{k,\ell-1} - \mathbf{u}_{k,\ell}\|_2$ using

$$\|\mathbf{v}_{k,\ell-1} - \mathbf{u}_{k,\ell}\|_2 \leq \|\mathbf{v}_{k,\ell-1} - \mathbf{u}_{k,\ell-1}\|_2 + \|\mathbf{u}_{k,\ell} - \mathbf{u}_{k,\ell-1}\|_2$$
$$\leq \|\mathbf{v}_{k,\ell-1} - \mathbf{u}_{k,\ell-1}\|_2 + \|\mathbf{u}_{k,\ell} - \mathbf{u}_k^\star\|_2 + \|\mathbf{u}_{k,\ell-1} - \mathbf{u}_k^\star\|_2$$

Combining this upper bound with (12) gives

$$\|\mathbf{v}_{k,\ell} - \mathbf{u}_{k,\ell}\|_2 \leq \mathcal{F}_k \left( \|\mathbf{v}_{k,\ell-1} - \mathbf{u}_{k,\ell-1}\|_2 + \|\mathbf{u}_{k,\ell} - \mathbf{u}_k^\star\|_2 + \|\mathbf{u}_{k,\ell-1} - \mathbf{u}_k^\star\|_2 \right) \tag{13}$$

Moreover, the triangle inequality implies that

$$\|\mathbf{v}_{k,\ell} - \mathbf{u}_k^\star\|_2 \leq \|\mathbf{v}_{k,\ell} - \mathbf{u}_{k,\ell}\|_2 + \|\mathbf{u}_{k,\ell} - \mathbf{u}_k^\star\|_2 \tag{14}$$

Now, (13) and (14) give a pretty good characterization of the propagation of the errors. It remains to characterize $\|\mathbf{u}_{k,\ell} - \mathbf{u}_k^\star\|_2$ for each $\ell$, and then we can dive into solving the recurrence. A naive bound would be that $\|\mathbf{u}_{k,\ell} - \mathbf{u}_k^\star\|_2 \leq 2$, as $\|\mathbf{u}_{k,\ell}\|_2 = \|\mathbf{u}_k^\star\|_2 = 1$. However, notice that $\mathbf{u}_{k,\ell}$ is the top eigenvector of $\mathbf{\Sigma}_{k,\ell}$ and $\mathbf{\Sigma}_k^\star$, respective. Thus, we can invoke the Davis-Kahan Theorem to obtain a tighter bound. This property is given by Lemma 2, whose proof is deferred to Appendix B.2.

**Lemma 2.** *Assume that* $1 = \lambda_1^\star > \lambda_2^\star > \ldots$ *. If the following inequality holds for some* $c_0 > 1$

$$\sum_{k'=1}^{k-1} \lambda_{k'}^\star \|\mathbf{u}_{k'}^\star - \mathbf{v}_{k',\ell-1}\|_2 \leq \frac{c_0 - 1}{4c_0} \left( \lambda_k^\star - \lambda_{k+1}^\star \right) \tag{15}$$

*then we have that*

$$\|\mathbf{u}_{k,\ell} - \mathbf{u}_k^\star\|_2 \leq \frac{4c_0}{\lambda_k^\star - \lambda_{k+1}^\star} \sum_{k'=1}^{k-1} \lambda_{k'}^\star \|\mathbf{u}_{k'}^\star - \mathbf{v}_{k',\ell-1}\|_2 \tag{16}$$

Now, we are going to use induction to proceed with the proof. Notice that, in order to control $\|\mathbf{u}_{k,\ell} - \mathbf{u}_k^\star\|_2$ using Lemma 2, one only need to control the recovery error of all previous eigenvectors $\|\mathbf{u}_{k'}^\star - \mathbf{v}_{k',\ell-1}\|_2$, as given in (15). Thus, fix some $k$, we will assume the inductive hypothesis that there exists some $s$ such that for all $\ell \geq s$, we can guarantee (15). For the case of $k = 1$, this is obvious, as the left-hand side of (15) is 0. When $k \geq 1$ and we can gather the conditions as

$$\|\mathbf{v}_{k,\ell} - \mathbf{u}_{k,\ell}\|_2 \leq \mathcal{F}_{k,\ell} \left( \|\mathbf{v}_{k,\ell-1} - \mathbf{u}_{k,\ell-1}\|_2 + \|\mathbf{u}_{k,\ell} - \mathbf{u}_k^\star\|_2 + \|\mathbf{u}_{k,\ell-1} - \mathbf{u}_k^\star\|_2 \right)$$
$$\|\mathbf{u}_k^\star - \mathbf{v}_{k,\ell}\|_2 \leq \|\mathbf{v}_{k,\ell} - \mathbf{u}_{k,\ell}\|_2 + \|\mathbf{u}_{k,\ell} - \mathbf{u}_k^\star\|_2$$
$$\|\mathbf{u}_{k,\ell} - \mathbf{u}_k^\star\|_2 \leq \frac{4c_0}{\lambda_k^\star - \lambda_{k+1}^\star} \sum_{k'=1}^{k-1} \lambda_{k'}^\star \|\mathbf{u}_{k'}^\star - \mathbf{v}_{k',\ell-1}\|_2 \, ; \quad \forall \ell \geq s_k$$

For simplicity, we let

$$\|\mathbf{v}_{k,\ell} - \mathbf{u}_{k,\ell}\|_2 =: D_{k,\ell}; \quad \|\mathbf{u}_{k,\ell} - \mathbf{u}_k^\star\|_2 =: B_{k,\ell}; \quad \|\mathbf{u}_k^\star - \mathbf{v}_{k,\ell}\|_2 =: G_{k,\ell}$$

Moreover, we let $\mathcal{C}_k = \frac{4c_0}{\lambda_k^\star - \lambda_{k+1}^\star}$. Then the iterates are simplified to

$$D_{k,\ell} \leq \mathcal{F}_k \left( D_{k,\ell-1} + B_{k,\ell} + B_{k,\ell-1} \right)$$
$$G_{k,\ell} \leq D_{k,\ell} + B_{k,\ell}$$
$$B_{k,\ell} \leq \mathcal{C}_k \sum_{k'=1}^{k-1} \lambda_{k'}^\star G_{k',\ell-1}$$

where we set $G_{0,\ell} = 0$ for all $\ell$. Then $G_{k,\ell}$ can be written as

$$G_{k,\ell} \leq \mathcal{F}_k^{\ell-s} D_{k,s} + \sum_{\ell'=s}^{\ell-1} \mathcal{F}_k^{\ell-\ell'} \left( B_{k,\ell'} + B_{k,\ell'-1} \right) + B_{k,\ell}$$

for any $s \in [\ell]$. Here, the first term can be made small as long as we choose a large enough $\ell$. The third term is the unavoidable error propagation. The second term can cause $G_{k,\ell}$ to grow, and needs a careful analysis. To understand the recurrence between $G_{k,\ell}$ and $B_{k,\ell}$, we use the following lemma

**Lemma 3.** *Let $\hat{s}_k$ be given for all $k \in [K]$ such that $1 \leq \hat{s}_1 \leq \cdots \leq \hat{s}_K$. Let $s_k \in \mathbb{Z}$ be given for all $k \in [K]$ such that $1 = s_0 \leq s_1 \leq \cdots \leq s_K$. Consider the sequence $\{B_{k,\ell}\}_{\ell=\hat{s}_k}^{\infty}$ and $\{G_{k,\ell}\}_{\ell=s_k-1}^{\infty}$ for all $k \in [K]$ characterized by the following recurrence*

$$B_{k,\ell} \leq C_k \sum_{k'=1}^{k-1} \lambda_{k'}^{\star} G_{k',\ell-1}$$

$$G_{k,\ell} \leq \mathcal{F}_k^{\ell-s_k+1} D_{k,s_k-1} + \sum_{\ell'=s_k-1}^{\ell-1} \mathcal{F}_k^{\ell-\ell'} \left(B_{k,\ell'} + B_{k,\ell'-1}\right) + B_{k,\ell} \tag{17}$$

*Let $m_k = \max\{\mathcal{F}_k, \gamma_{k-1}\}$ for all $k \in [K]$ and $m_0 = -1$. Let $\{\gamma\}_{k=-1}^{K}$ be given such that $\gamma_{-1} = \gamma_0 = 0$ and $\gamma_k = \frac{1}{k+1} + \frac{k}{k+1} m_k$ for all $k \in [K]$. Define sequences $\{\hat{B}_{k,\ell}\}_{\ell=\hat{s}_k}^{\infty}$ and $\{\hat{G}_{k,\ell}\}_{\ell=s_k-1}^{\infty}$ for all $k \in [K]$ as*

$$\hat{B}_{k,\ell} = \begin{cases} \min\left\{2, m_{k-1}^{\ell-\hat{s}_k}(\ell - \hat{s}_k + 1)\hat{B}_{k,\hat{s}_k}\right\} & \text{if } \ell > \hat{s}_k \\ C_k \sum_{k'=1}^{k-1} \lambda_{k'}^{\star} \hat{G}_{k',\hat{s}_k-1} & \text{if } \ell = \hat{s}_k \end{cases}$$

$$\hat{G}_{k,\ell} = \begin{cases} m_k^{\ell-s_k+1}(\ell - s_k + 2)\hat{G}_{k,s_k-1} & \text{if } \ell \geq s_k \\ D_{k,s_k-1} + \hat{B}_{k,s_k-1} + \hat{B}_{k,s_k-2} & \text{if } \ell = s_k - 1 \end{cases} \tag{18}$$

*Suppose that $\hat{s}_{k+1} \geq s_k$, and $s_k$ satisfies satisfies $m_{k-1}^{s_k-\hat{s}_k-2} \leq \frac{1}{s_k-\hat{s}_k-1}$ and $s_k \geq \frac{km_{k-1}}{1-m_{k-1}} + \hat{s}_k + 2$. Moreover, suppose that $\hat{B}_{k,\hat{s}_k} \leq 2$ for all $k \in [K]$. Then the following two conditions hold*

1. $\hat{B}_{k,\ell} \geq B_{k,\ell}$ *for all* $\ell \geq \hat{s}_k$

2. $\hat{G}_{k,\ell} \geq G_{k,\ell}$ *for all* $\ell \geq s_k - 1$

The proof of Lemma 3 is deferred to Appendix B.3. Lemma 3 implies that under proper condition of $s_k$ and $\hat{s}_k$, we have

$$\begin{aligned} G_{k,\ell} &\leq \hat{G}_{k,\ell} \\ &\leq \max\{\mathcal{F}_k, \gamma_{k-1}\}^{\ell-s_k+1}(\ell - s_k + 1)\hat{G}_{k,s_k-1} \\ &= \max\{\mathcal{F}_k, \gamma_{k-1}\}^{\ell-s_k+1}(\ell - s_k + 1)\left(D_{k,s_k-1} + \hat{B}_{k,s_k-1} + \hat{B}_{k,s_k-2}\right) \end{aligned}$$

By definition, we have that $D_{k,s_k-1} = \|\mathbf{v}_{k,s_k-1} - \mathbf{u}_{k,s_k-1}\|_2 \leq 2$. Moreover, the definition in (18) gives that $\hat{B}_{k,s_k-1} \leq 2$ and $\hat{B}_{k,s_k-2} \leq 2$. Therefore, we can conclude that

$$G_{k,\ell} \leq 6(\ell - s_k + 1)\max\{\mathcal{F}_k, \gamma_{k-1}\}^{\ell-s_k+1}$$

Now, we go back to the condition of $s_k$ and $\hat{s}_k$. The requirement of $\{s_k\}_{k=0}^{K}$ and $\{\hat{s}_k\}_{k=1}^{K}$ can be gathered below

1. $1 = s_0 \leq s_1 \leq \ldots s_K$ and $1 \leq \hat{s}_1 \leq \cdots \leq \hat{s}_K$

2. $\hat{s}_{k+1} \geq s_k$ and $s_k \geq \frac{km_{k-1}}{1-m_{k-1}} + \hat{s}_k + 2$

3. $m_{k-1}^{s_k} - \hat{s}_k - 2 \leq \frac{1}{s_k-\hat{s}_k-1}$

4. $m_{k-1}^{\ell-\hat{s}_k}(\ell - \hat{s}_k + 1)\sum_{k'=1}^{k-1} \lambda_{k'}^{\star} \hat{G}_{k',\hat{s}_k-1} \leq \frac{c_0-1}{4c_0}\left(\lambda_k^{\star} - \lambda_{k+1}^{\star}\right)$ for all $\ell \geq \hat{s}_k$

where the first three conditions are directly required by Lemma 3, and the fourth condition is required because the upper bound on $B_{k,\ell}$ in (17) hold only when

$$\sum_{k'=1}^{k-1} \lambda_{k'}^{\star} \|\mathbf{u}_{k'}^{\star} - \mathbf{v}_{k',\ell-1}\|_2 \leq \frac{c_0-1}{4c_0}\left(\lambda_k^{\star} - \lambda_{k+1}^{\star}\right)$$

from Lemma 2. Notice that since $\hat{B}_{k,\hat{s}_k} = C_k \sum_{k'=1}^{k-1} \lambda_{k'}^{\star} \hat{G}_{k',\hat{s}_k-1}$, enforcing the fourth condition directly implies that $\hat{B}_{k,\hat{s}_k} \leq 2$. Now, we are going to simplify these conditions. A useful tool will be the following lemma, whose proof is provided in Appendix B.4.

**Lemma 4.** *Let $m \in (0, 1)$ and $\epsilon \in \mathbb{R}$ be given. Let $g(x) = m^x(x + 1)$, and let $W_{-1}(\cdot)$ be the Lambert-W function. Then*

    *1. When $\epsilon \geq -\frac{1}{em \log m}$, then any $x \geq 0$ satisfies $g(x) \leq \epsilon$*

    *2. When $\epsilon \leq -\frac{1}{em \log m}$, then any $x \geq \frac{1}{\log m} W_{-1}(\epsilon m \log m) - 1$ satisfies $g(x) \leq \epsilon$*

Notice that #4 in the conditions above implies that $\|\mathbf{\Sigma}_{k,\ell} - \mathbf{\Sigma}_k^\star\|_F \leq \frac{c_0 - 1}{c_0} (\lambda_k^\star - \lambda_{k+1}^\star)$ and

$$B_{k,\hat{s}_k} = \sum_{k'=1}^{k-1} \lambda_{k'}^\star \hat{G}_{k',\hat{s}_k - 1} \leq c_0 - 1$$

Choose $c_0 = 3$ then guarantees that $B_{k,\hat{s}_k} \leq 2$. Now, we aim at simplifying Condition #4 above. To start, we notice that the term $m_{k-1}^{\ell - \hat{s}_k}(\ell - \hat{s}_k + 1)$ achieves global maximum at $\ell - \hat{s}_k = \frac{1}{\log 1/m_{k-1}} - 1$ with value $\frac{1}{\log 1/m_{k-1}} m_{k-1}^{\frac{1}{\log 1/m_{k-1}} - 1}$. Therefore, it suffices to guarantee that

$$\sum_{k'=1}^{k-1} \lambda_{k'}^\star \hat{G}_{k',\hat{s}_k - 1} \leq -\log m_{k-1} \cdot m_{k-1}^{\frac{1}{\log m_{k-1}} - 1} \cdot \frac{1}{6} (\lambda_k^\star - \lambda_{k+1}^\star)$$

From Lemma 3, we have that for $\ell \geq s_k - 1$, $G_{k,\ell} \leq \hat{G}_{k,\ell}$, and

$$\hat{G}_{k,\ell} = m_k^{\ell - s_k + 1}(\ell - s_k + 2) \hat{G}_{k,s_k - 1}$$

with $\hat{G}_{k,s_k - 1} \leq 6$. Therefore, it suffices to guarantee that

$$\sum_{k'=1}^{k-1} \lambda_{k'}^\star m_{k'}^{\hat{s}_k - s_{k'} + 1}(\hat{s}_k - s_{k'} + 2) \hat{G}_{k,s_{k'} - 1} \leq -\log m_{k-1} \cdot m_{k-1}^{\frac{1}{\log m_{k-1}} - 1} \cdot \frac{1}{6} (\lambda_k^\star - \lambda_{k+1}^\star)$$

which would be satisfied if we have

$$\lambda_{k'}^\star m_{k'}^{\hat{s}_k - s_{k'} + 1}(\hat{s}_k - s_{k'} + 2) \hat{G}_{k,s_{k'} - 1} \leq -\frac{\lambda_k^\star - \lambda_{k+1}^\star}{6(k-1)} \log m_{k-1} \cdot m_{k-1}^{\frac{1}{\log m_{k-1}} - 1}$$

Thus, $\hat{s}_k$ must satisfy for all $s_{k'}$

$$m_{k'}^{\hat{s}_k - s_{k'} + 1}(\hat{s}_k - s_{k'} + 2) \leq -\frac{\lambda_k^\star - \lambda_{k+1}^\star}{36\lambda_{k'}^\star(k-1)} \log m_{k-1} \cdot m_{k-1}^{\frac{1}{\log m_{k-1}} - 1} \tag{19}$$

With the help of Lemma 4, Condition #3 transfers to

$$s_k \geq \frac{1}{\log m_{k-1}} W_{-1}(m_{k-1} \log m_{k-1}) + \hat{s}_k + 1$$

Similarly, Condition #4 transfers to

$$\hat{s}_k \geq \frac{1}{\log m_{k'}} W_{-1}\left(-\frac{\lambda_k^\star - \lambda_{k+1}^\star}{36\lambda_{k'}^\star(k-1)} \log m_{k-1} \cdot m_{k-1}^{\frac{1}{\log m_{k-1}} - 1} m_{k'} \log m_{k'}\right) + s_{k'} - 2$$

which can be guaranteed as long as

$$\hat{s}_k \geq \frac{1}{\log m_{k'}} W_{-1}\left(-\frac{\lambda_k^\star - \lambda_{k+1}^\star}{36k\lambda_{k'}^\star} (\log m_{k-1})^2 \cdot m_{k-1}^{\frac{1}{\log m_{k-1}}}\right) + s_{k'} - 2$$

Gathering all requirements, we have

$$s_k \geq \max\left\{\frac{1}{\log m_{k-1}} W_{-1}(m_{k-1} \log m_{k-1}), \frac{(k-1)m_{k-1} + 1}{1 - m_{k-1}}\right\} + \hat{s}_k + 1$$

$$\hat{s}_k \geq \frac{1}{\log m_{k'}} W_{-1}\left(-\frac{\lambda_k^\star - \lambda_{k+1}^\star}{36k\lambda_{k'}^\star} (\log m_{k-1})^2 \cdot m_{k-1}^{\frac{1}{\log m_{k-1}}}\right) + s_{k'} - 2$$

Plugging $\hat{s}_k$ into the lower bound of $s_k$ shows that the condition in (7) suffice to guarantee that Lemma 3 holds. Thus, we can conclude that

$$\|\mathbf{v}_{k,\ell} - \mathbf{u}_k^\star\|_2 = G_{k,\ell} \leq 6(\ell - s_k + 2) m_k^{\ell - s_k + 1}$$

which finishes the proof.

### B.2 PROOF OF LEMMA 2

Applying the Davis-Kahan Theorem, if we let $\lambda_{k+1,\ell} = \lambda_{\max}(\boldsymbol{\Sigma}_{k,\ell})$, then for all $k, \ell$ such that $\|\boldsymbol{\Sigma}_k^\star - \boldsymbol{\Sigma}_{k,\ell}\|_F < \lambda_k^\star - \lambda_{k+1}^\star$, we have

$$\|\mathbf{u}_{k,\ell} - \mathbf{u}_k^\star\|_2 \le \frac{\|\boldsymbol{\Sigma}_k^\star - \boldsymbol{\Sigma}_{k,\ell}\|_F}{\lambda_k^\star - \lambda_{k+1,\ell}}$$

By definition, we have

$$\boldsymbol{\Sigma}_k^\star = \boldsymbol{\Sigma} - \sum_{k'=1}^{k-1} \lambda_{k'}^\star \mathbf{u}_{k'}^\star \mathbf{u}_{k'}^{\star\top}; \quad \boldsymbol{\Sigma}_{k,\ell} = \boldsymbol{\Sigma} - \sum_{k'=1}^{k-1} \left(\mathbf{v}_{k',\ell-1}^\top \boldsymbol{\Sigma} \mathbf{v}_{k',\ell-1}\right) \mathbf{v}_{k',\ell-1} \mathbf{v}_{k',\ell-1}^\top$$

Thus, we can write the difference between the two matrices as

$$\boldsymbol{\Sigma}_k^\star - \boldsymbol{\Sigma}_{k,\ell} = \sum_{k'=1}^{k-1} \left(\lambda_{k'}^\star - \mathbf{v}_{k',\ell-1}^\top \boldsymbol{\Sigma} \mathbf{v}_{k',\ell-1}\right) \mathbf{v}_{k',\ell-1} \mathbf{v}_{k',\ell-1}^\top + \sum_{k'=1}^{k-1} \lambda_{k'}^\star \left(\mathbf{v}_{k',\ell-1} \mathbf{v}_{k',\ell-1}^\top - \mathbf{u}_{k'}^\star \mathbf{u}_{k'}^{\star\top}\right)$$

It is easy to see that for $\mathbf{v}_{k',\ell-1}$ and $\mathbf{v}_{k'}^\star$ with unit norm,

$$\left\|\mathbf{v}_{k',\ell-1} \mathbf{v}_{k',\ell-1}^\top - \mathbf{u}_{k'}^\star \mathbf{u}_{k'}^{\star\top}\right\|_2^2 = 2 - 2\langle \mathbf{v}_{k',\ell-1}, \mathbf{u}_{k'}^\star\rangle^2 \le \|\mathbf{u}_{k'}^\star - \mathbf{v}_{k',\ell-1}\|_2^2$$

Moreover to bound $\left|\lambda_{k'}^\star - \mathbf{v}_{k',\ell-1}^\top \boldsymbol{\Sigma} \mathbf{v}_{k',\ell-1}\right|$, we denote $\boldsymbol{\delta} = \mathbf{v}_{k',\ell-1} - \mathbf{u}_{k'}^\star$, and write

$$\mathbf{v}_{k',\ell-1}^\top \boldsymbol{\Sigma} \mathbf{v}_{k',\ell-1} = \left(\mathbf{u}_{k'}^\star - \boldsymbol{\delta}\right)^\top \boldsymbol{\Sigma} \left(\mathbf{u}_{k'}^\star - \boldsymbol{\delta}\right) = \lambda_{k'}^\star - 2\lambda_{k'}^\star \boldsymbol{\delta}^\top \mathbf{u}_{k'}^\star + \boldsymbol{\delta}^\top \boldsymbol{\Sigma} \boldsymbol{\delta}$$

Therefore, we have

$$\left|\lambda_{k'}^\star - \mathbf{v}_{k',\ell-1}^\top \boldsymbol{\Sigma} \mathbf{v}_{k',\ell-1}\right| = \left|-2\lambda_{k'} \boldsymbol{\delta}^\top \mathbf{u}_{k'}^\star + \boldsymbol{\delta}^\top \boldsymbol{\Sigma} \boldsymbol{\delta}\right| \le 2\lambda_{k'}^\star \|\mathbf{v}_{k',\ell-1} - \mathbf{u}_{k'}^\star\|_2 + \lambda_1^\star \|\mathbf{v}_{k',\ell-1} - \mathbf{u}_{k'}^\star\|_2^2$$

This gives

$$\|\boldsymbol{\Sigma}_k^\star - \boldsymbol{\Sigma}_{k,\ell}\|_F \le \sum_{k'=1}^{k-1} \left(3\lambda_{k'}^\star \|\mathbf{u}_{k'}^\star - \mathbf{v}_{k',\ell-1}\|_2 + \lambda_1^\star \|\mathbf{u}_{k'}^\star - \mathbf{v}_{k',\ell-1}\|_2^2\right)$$

We then need to assume that, for some $c_0 > 1$,

$$\sum_{k'=1}^{k-1} \lambda_{k'}^\star \|\mathbf{u}_{k'}^\star - \mathbf{v}_{k',\ell-1}\|_2 \le \frac{c_0 - 1}{4c_0}\left(\lambda_k^\star - \lambda_{k+1}^\star\right)$$

In this scenario, we can conclude that $\|\mathbf{u}_{k'}^\star - \mathbf{v}_{k',\ell-1}\|_2 \le \lambda_k^\star$. Combined with the condition that $\lambda_1^\star = 1$, we have

$$\|\boldsymbol{\Sigma}_k^\star - \boldsymbol{\Sigma}_{k,\ell}\|_F \le 4 \sum_{k'=1}^{k-1} \lambda_{k'}^\star \|\mathbf{u}_{k'}^\star - \mathbf{v}_{k',\ell-1}\|_2 \le \frac{c_0 - 1}{c_0}\left(\lambda_k^\star - \lambda_{k+1}^\star\right)$$

Moreover, we have

$$\|\mathbf{u}_{k,\ell} - \mathbf{u}_k^\star\|_2 \le \frac{4}{\lambda_k^\star - \lambda_{k+1,\ell}} \sum_{k'=1}^{k-1} \lambda_{k'}^\star \|\mathbf{u}_{k'}^\star - \mathbf{v}_{k',\ell-1}\|_2 \le \frac{4c_0}{\lambda_k^\star - \lambda_{k+1}^\star} \sum_{k'=1}^{k-1} \lambda_{k'}^\star \|\mathbf{u}_{k'}^\star - \mathbf{v}_{k',\ell-1}\|_2$$

where the last inequality follows from $\lambda_k^\star - \lambda_{k+1,\ell} \ge \lambda_k^\star - \lambda_{k+1}^\star - \|\boldsymbol{\Sigma}_k^\star - \boldsymbol{\Sigma}_{k,\ell}\|_F \ge \frac{1}{c_0}\left(\lambda_k^\star - \lambda_{k+1}^\star\right)$

### B.3 PROOF OF LEMMA 3

To start, we will need to prove an auxiliary lemma

**Lemma 5.** *Let the sequence $\{\hat{B}_{k,\ell}\}_{\ell=s_{k-1}+1}^\infty$ be defined as*

$$\hat{B}_{k,\ell} = \min\left\{2, m_{k-1}^{\ell-\hat{s}_k}\left(\ell - \hat{s}_k + 1\right)\hat{B}_{k,\hat{s}_k}\right\}$$

*with some $\hat{B}_{k,\hat{s}_k} \le 2$ and $m_{k-1} \in (0,1)$. Then for all $s$ that satisfies $m_{k-1}^{s-\hat{s}_k} \le \frac{1}{s-\hat{s}_k+1}$ and $s \ge \frac{km_{k-1}}{1-m_{k-1}} + \hat{s}_k$, we have that*

$$\hat{B}_{k,\ell} \le \left(\frac{1}{k} + \frac{k-1}{k}m_{k-1}\right)^{\ell-s} \hat{B}_{k,s}$$

*Proof.* To start, by definition, we can write $\hat{B}_{k,s}$ as

$$\hat{B}_{k,s} = \min\left\{2, m_{k-1}^{s-\hat{s}_k}(s - \hat{s}_k + 1)\hat{B}_{k,\hat{s}_k}\right\}$$

Since $\hat{B}_{k,\hat{s}_k} \leq 2$, we have

$$m_{k-1}^{s-\hat{s}_k}(s - \hat{s}_k + 1)\hat{B}_{k,\hat{s}_k} \leq 2m_{k-1}^{s-\hat{s}_k}(s - \hat{s}_k + 1) \leq 2$$

where the last inequality follows from the condition $m_{k-1}^{s-\hat{s}_k} \leq \frac{1}{s-\hat{s}_k+1}$. Therefore, we can write $\hat{B}_{k,s}$ as

$$\hat{B}_{k,s} = m_{k-1}^{s-\hat{s}_k}(s - \hat{s}_k + 1)\hat{B}_{k,\hat{s}_k}$$

Recall that for any $\ell \geq s$ we have

$$\hat{B}_{k,\ell} = \min\left\{2, m_{k-1}^{\ell-\hat{s}_k}(\ell - \hat{s}_k + 1)\hat{B}_{k,\hat{s}_k}\right\}$$

Plugging in $\hat{B}_{k,\hat{s}_k} = \left(m_{k-1}^{s-\hat{s}_k}(s - \hat{s}_k + 1)\right)^{-1}\hat{B}_{k,s}$ we have

$$\hat{B}_{k,\ell} \leq m_{k-1}^{\ell-s} \cdot \frac{\ell - \hat{s}_k + 1}{s - \hat{s}_k + 1} \cdot \hat{B}_{k,s}$$

$$\leq m_k^{\ell-s}\left(\prod_{\ell'=s+1}^{\ell} \frac{\ell' - \hat{s}_k + 1}{\ell' - \hat{s}_k}\right) \cdot \hat{B}_{k,s}$$

$$\leq m_k^{\ell-s}\left(\frac{s - \hat{s}_k + 1}{s - \hat{s}_k}\right)^{\ell-s}\hat{B}_{k,s}$$

$$= \left(m_k \cdot \frac{s - \hat{s}_k + 1}{s - \hat{s}_k}\right)^{\ell-s}\hat{B}_{k,s}$$

$$\leq \left(\frac{1}{k} + \frac{k-1}{k}m_k\right)^{\ell-s}\hat{B}_{k,s}$$

where the last inequality is because $s \geq \frac{km_k}{1-m_k} + \hat{s}_k$ implies that

$$m_k \cdot \frac{s - \hat{s}_k + 1}{s - \hat{s}_k} \leq m_k \cdot \frac{\frac{km_k}{1-m_k} + 1}{\frac{km_k}{1-m_k}} = \frac{(k-1)m_k + 1}{k} = \frac{1}{k} + \frac{k-1}{k}m_k$$

This completes the proof. $\qquad\square$

We will use induction on $k$ to prove the lemma.

**Base Case: $k = 1$.** In this case, by the definition of $\hat{B}_{1,\hat{s}_1}$, we have $\hat{B}_{1,s_1} = 0$. Moreover, by the definition of $\hat{B}_{1,\ell}$ for $\ell \geq s_1$, we have $\hat{B}_{1,\ell} = m_0^{\ell-\hat{s}_k}(\ell - \hat{s}_k + 1)\hat{B}_{k,\hat{s}_k} = 0$. Lastly, by the definition of $B_{1,\ell}$, we have $B_{1,\ell} = 0$. Therefore, we must have that $\hat{B}_{1,\ell} = 0 = B_{1,\ell}$ for all $\ell \geq \hat{s}_1$. This shows Condition #1. Using $\hat{B}_{1,\ell} = 0 = B_{1,\ell}$, we can derive that $\hat{G}_{1,\ell} = \mathcal{F}_1^{\ell-s_1+1}D_{1,s_1-1}$, and $G_{1,\ell} = \mathcal{F}_1^{\ell-s_1+1}D_{1,s_1-1}$. This implies that $G_{1,\ell} \leq \hat{G}_{1,\ell}$, and shows Condition #2. Thus, we have shown that the case $k = 1$ holds.

**Inductive Step.** Now, we assume that for all $\hat{k} \leq k$, the following holds

1. $\hat{B}_{\hat{k},\ell} \geq B_{\hat{k},\ell}$ for all $\ell \geq \hat{s}_{\hat{k}}$

2. $\hat{G}_{\hat{k},\ell} \geq G_{\hat{k},\ell}$ for all $\ell \geq s_{\hat{k}} - 1$

We wish to show that the above three conditions hold for $\hat{k} = k + 1$. We start by showing Condition #1 for $\hat{k} = k + 1$. By Condition #2 in the inductive hypothesis, we have that $\hat{G}_{\hat{k},\ell} \geq G_{\hat{k},\ell}$ for all

$\ell \geq s_{\hat{k}} - 1$. Since $\hat{s}_{k+1} \geq s_{\hat{k}}$ for all $\hat{k} \leq k$, we have that $\hat{G}_{\hat{k},\ell} \geq G_{\hat{k},\ell}$ for all $\ell \geq \hat{s}_{k+1} - 1$. Therefore, in the case of $\ell = \hat{s}_{k+1}$

$$B_{k+1,\ell} \leq C_{k+1} \sum_{k'=1}^{k} \lambda_{k'}^{\star} G_{k',\ell-1} \leq C_{k+1} \sum_{k'=1}^{k} \lambda_{k'}^{\star} \hat{G}_{k',\ell-1} = \hat{B}_{k,\ell}$$

Next, we show that $\hat{B}_{k+1,\ell} \geq B_{k+1,\ell}$ for all $\ell \geq \hat{s}_{k+1}$. If $\hat{B}_{k+1,\ell} \geq 2$, then we directly have $\hat{B}_{k+1,\ell} \geq B_{k+1,\ell}$ since $B_{k+1,\ell} \leq 2$. Otherwise, suppose $\hat{B}_{k+1,\ell} \leq 2$. Since $\hat{s}_{k+1} \geq s_{k'}$ for all $k' \leq k$, by the definition of $\hat{G}_{k,\ell}$, we have

$$\hat{G}_{k',\hat{s}_{k+1}-1} = m_{k'}^{\hat{s}_{k+1}-s_{k'}} (\hat{s}_{k+1} - s_{k'} + 1) \hat{G}_{k',s_{k'}-1}$$

Based on the definition of $\hat{B}_{k+1,s_k}$, and since $m_k \geq m_{k'}$ for all $k \geq k$, we have that

$$\hat{B}_{k+1,\ell} = m_k^{\ell-\hat{s}_{k+1}} (\ell - \hat{s}_{k+1} + 1) \hat{B}_{k+1,\hat{s}_{k+1}}$$

$$= m_k^{\ell-\hat{s}_{k+1}} (\ell - \hat{s}_{k+1} + 1) C_{k+1} \sum_{k'=1}^{k} \lambda_{k'}^{\star} \hat{G}_{k',\hat{s}_{k+1}-1}$$

$$\geq C_{k+1} \sum_{k'=1}^{k} \lambda_{k'}^{\star} m_{k'}^{\ell-\hat{s}_{k+1}} (\ell - \hat{s}_{k+1} + 1) \hat{G}_{k',\hat{s}_{k+1}-1}$$

$$\geq C_{k+1} \sum_{k'=1}^{k} \lambda_{k'}^{\star} m_{k'}^{\ell-s_{k'}} (\ell - \hat{s}_{k+1} + 1) (s_{k+1} - s_{k'} + 1) \hat{G}_{k',s_{k'}-1}$$

$$\geq C_{k+1} \sum_{k'=1}^{k} \lambda_{k'}^{\star} m_{k'}^{\ell-s_{k'}} (\ell - s_{k'} + 1) \hat{G}_{k',s_{k'}-1}$$

$$= C_{k+1} \sum_{k'=1}^{k} \lambda_{k'}^{\star} \hat{G}_{k',\ell-1}$$

where the third to the last inequality is due to $(\ell - \hat{s}_{k+1} + 1) + (s_{k+1} - s_{k'} + 1) - 1 = \ell - s_{k'} + 1$, and for all $a \geq 1, b \geq 1$, we will have $ab \geq a + b - 1$. By the inductive hypothesis, we have that $\hat{G}_{k',\ell} \geq G_{k',\ell}$ for all $\ell \geq \hat{s}_{k+1} \geq s_{k'} - 1$. Therefore, it must hold that

$$\hat{B}_{k+1,\ell} \geq C_{k+1} \sum_{k'=1}^{k} \lambda_{k'}^{\star} G_{k',\ell-1} \geq B_{k+1,\ell}$$

This proves Condition #1 for $\hat{k} = k + 1$. Next, we will prove Condition #2 for $\hat{k} = k + 1$. To start, when $\ell = s_{k+1} - 1$, we have

$$\hat{G}_{k+1,\ell} = D_{k+1,s_{k+1}-1} + \hat{B}_{k+1,s_{k+1}-1} + \hat{B}_{k+1,s_{k+1}-2}$$

while by (17) we have

$$G_{k+1,\ell} \leq D_{k+1,s_{k+1}-1} + B_{k+1,s_{k+1}-1}$$

Since $\hat{B}_{k+1,s_{k+1}-2} \geq 0$ and $\hat{B}_{k+1,s_{k+1}-1} \geq B_{k+1,s_{k+1}-1}$ as proved above for $s_{k+1} \geq \hat{s}_{k+1} - 2$, we must have that $\hat{G}_{k+1,\ell} \geq G_{k+1,\ell}$ when $\ell = s_{k+1} - 1$. Next, we show that $\hat{G}_{k+1,\ell} \geq G_{k+1,\ell}$ when $\ell > s_{k+1} - 1$. To start,

$$G_{k+1,\ell} \leq \mathcal{F}_{k+1}^{\ell-s_{k+1}+1} D_{k+1,s_{k+1}-1} + \sum_{\ell'=s_{k+1}-1}^{\ell-1} \mathcal{F}_{k+1}^{\ell-\ell'} (B_{k+1,\ell'} + B_{k+1,\ell'-1}) + B_{k+1,\ell}$$

$$\leq \mathcal{F}_{k+1}^{\ell-s_{k+1}+1} D_{k+1,s_{k+1}-1} + \sum_{\ell'=s_{k+1}-1}^{\ell} \mathcal{F}_{k+1}^{\ell-\ell'} (B_{k+1,\ell'} + B_{k+1,\ell'-1})$$

$$\leq \mathcal{F}_{k+1}^{\ell-s_{k+1}+1} D_{k+1,s_{k+1}-1} + \sum_{\ell'=s_{k+1}-1}^{\ell} \mathcal{F}_{k+1}^{\ell-\ell'} \left( \hat{B}_{k+1,\ell'} + \hat{B}_{k+1,\ell'-1} \right)$$

By definition of $\hat{B}_{k+1,\ell}$, invoking Lemma 5 with $s = s_{k+1} - 2$ and $s = s_{k+1} - 1$, we have that, as long as $s_{k+1}$ satisfies $m_k^{s_{k+1}-\hat{s}_{k+1}-2} \leq \frac{1}{s_{k+1}-\hat{s}_{k+1}-1}$ and $s_{k+1} \geq \frac{(k+1)m_k}{1-m_k} + \hat{s}_{k+1} + 2$, it holds that

$$\hat{B}_{k+1,\ell} \leq \gamma_k^{\ell-s_{k+1}+2} \hat{B}_{k+1,s_{k+1}-2}$$
$$\hat{B}_{k+1,\ell} \leq \gamma_k^{\ell-s_{k+1}+1} \hat{B}_{k+1,s_{k+1}-1}$$

for all $\ell \geq s_k$. Therefore

$$G_{k+1,\ell} \leq \mathcal{F}_{k+1}^{\ell-s_{k+1}+1} D_{k+1,s_{k+1}-1} + \sum_{\ell'=s_{k+1}-1}^{\ell} \mathcal{F}_{k+1}^{\ell-\ell'} \left( \hat{B}_{k+1,\ell'} + \hat{B}_{k+1,\ell'-1} \right)$$

$$= \mathcal{F}_{k+1}^{\ell-s_{k+1}+1} D_{k+1,s_{k+1}-1} + \sum_{\ell'=s_{k+1}-1}^{\ell} \mathcal{F}_{k+1}^{\ell-\ell'} \gamma_k^{\ell'-s_{k+1}+1} \left( \hat{B}_{k+1,s_{k+1}-1} + \hat{B}_{k+1,s_{k+1}-2} \right)$$

$$\leq \max\{\mathcal{F}_{k+1}, \gamma_k\}^{\ell-s_{k+1}+1} \left( D_{k+1,s_{k+1}-1} + (\ell - s_{k+1} + 1) \left( \hat{B}_{k+1,s_{k+1}-1} + \hat{B}_{k+1,s_{k+1}-2} \right) \right)$$

$$\leq \max\{\mathcal{F}_{k+1}, \gamma_k\}^{\ell-s_{k+1}+1} (\ell - s_{k+1} + 1) \left( D_{k+1,s_{k+1}-1} + \hat{B}_{k+1,s_{k+1}-1} + \hat{B}_{k+1,s_{k+1}-2} \right)$$

$$= \hat{G}_{k+1,\ell}$$

where in the last equality we use $m_{k+1} = \max\{\mathcal{F}_{k+1}, \gamma_k\}$ and

$$\hat{G}_{k+1,s_{k+1}-1} = D_{k+1,s_{k+1}-1} + \hat{B}_{k+1,s_{k+1}-1} + \hat{B}_{k+1,s_{k+1}-2}$$

This proves Condition #2 under $\ell > s_{k+1} - 1$, which finishes the induction step and completes the proof.

### B.4 PROOF OF LEMMA 4

First, we prove the case $\epsilon \geq -\frac{m}{e \log m}$. Notice that the function $g(x)$ achieves global maximum at $x = \frac{1}{\log 1/m} - 1$ with value $\frac{1}{\log 1/m} m^{\frac{1}{\log 1/m} - 1}$. Moreover, notice that

$$\frac{1}{\log 1/m} m^{\frac{1}{\log 1/m} - 1} = -\frac{m^{\frac{1}{-\log m}}}{m \log m} = -\frac{e^{-\frac{1}{\log m} \cdot \log m}}{m \log m} = -\frac{1}{em \log m} \leq \epsilon$$

Therefore, for all $x \geq 0$ we would have $g(x) \leq \epsilon$. Next, we consider the case $\epsilon \leq -\frac{1}{em \log m}$. In this case, $x \geq \frac{1}{\log m} W_{-1}(\epsilon m \log m) - 1$ implies that

$$(x + 1) \log m \leq W_{-1}(\epsilon m \log m)$$

By the monotonicity of $W_{-1}$, we have

$$(x + 1) \log m \cdot e^{(x+1) \log m} \geq \epsilon m \log m$$

which gives $(x + 1) e^{x \log m} \leq \epsilon$. Thus, we have $g(x) = (x + 1)m^x \leq \epsilon$.

## C  Stochastic Parallel Deflation Algorithm

In this section, we provide the explicit form of the stochastic version of the parallel deflation algorithm as discussed in Section 3. Notice that in this algorithm we choose Hebb's rule as the $\texttt{Top} - 1$ subroutine for the convenience of a clearer presentation. However, any subroutine that use $\Sigma_{k,\ell}$ only for a matrix-vector multiplication can enjoy a similar efficient implementation.

---

**Algorithm 2** Stochastic Parallel Deflation with Hebb's Rule

---

**Require:** Batch of data in the $(\ell, t)$th iteration $\hat{\mathbf{Y}}_{\ell,t}$; # of eigenvectors (workers) $K$; # of iterations $T$; global communication rounds $L \geq K$, step size $\eta$.
**Ensure:** Approximate eigenvectors $\{\mathbf{v}_k\}_{k=1}^{K}$.
1: **for** $k = 1, \ldots, K$ **do**
2:     Randomly initialize $\hat{\mathbf{v}}_{k,\texttt{init}}$ with unit norm;
3: **end for**
4: **for** $\ell = 1, \ldots, L$ **do**
5:     **for** $k = 1, \ldots, K$ **do**
6:         **if** $k \leq \ell$ **then**
7:             Receive $\mathbf{v}_{1,\ell-1}, \ldots, \mathbf{v}_{k-1,\ell-1}$
8:             $\mathbf{v}_{k,\ell,0} := \mathbf{v}_{k,\ell-1}$;
9:             **for** $t = 1, \ldots, T$ **do**
10:                 $\hat{\lambda}_{k',\ell,t} = \|\hat{\mathbf{Y}}_{\ell,t}\mathbf{v}_{k',\ell-1}\|_2^2 \quad \forall k' \in [k-1]$;
11:                 $\mathbf{g}_{k,\ell,t} = \hat{\mathbf{Y}}_{k,\ell,t}^\top \hat{\mathbf{Y}}_{k,\ell,t}\mathbf{v}_{k,\ell,t-1} - \sum_{k'=1}^{k-1} \hat{\lambda}_{k',\ell,t} \left(\mathbf{v}_{k',\ell-1}^\top \mathbf{v}_{k,\ell,t-1}\right) \cdot \mathbf{v}_{k',\ell-1}$
12:                 $\mathbf{v}_{k,\ell,t} := (\mathbf{v}_{k,\ell,t-1} - \eta\mathbf{g}_{k,t,\ell})/\|\mathbf{v}_{k,\ell,t-1} - \eta\mathbf{g}_{k,t,\ell}\|_2$
13:             **end for**
14:             Broadcast $\mathbf{v}_{k,\ell} := \mathbf{v}_{k,\ell,T}$
15:         **else**
16:             $\mathbf{v}_{k,\ell} := \hat{\mathbf{v}}_{k,\texttt{init}}$;
17:         **end if**
18:     **end for**
19: **end for**
20: **return** $\{\mathbf{v}_{k,L}\}_{k=1}^{K}$

---

## D  Baseline Algorithms

We provide the generalization of the EigenGame-$\alpha$ Gemp et al. (2020) and EigenGame-$\mu$ Gemp et al. (2022) algorithms with multiple iterations of local updates $T \geq 1$ in Algorithm 3 and Algorithm 4. In particular, it should be noted that EigenGame-$\alpha$ and EigenGame-$\mu$ use covariance matrices computed on subsets of the data in each iteration, where in our case we assume that the covariance matrix is computed on the whole dataset before the algorithm runs. Moreover, if we set $T = 1$ in both Algorithm 3 and Algorithm 4, then we recover the original EigenGame-$\alpha$ and EigenGame-$\mu$ algorithms.

---

**Algorithm 3** EigenGame-$\alpha$

---

**Require:** $\Sigma \in \mathbb{R}^{d \times d}$; # of eigenvectors (workers) $K$; # of iterations $T$; global communication rounds $L \geq K$, step size $\eta$.
**Ensure:** Approximate eigenvectors $\{\mathbf{v}_k\}_{k=1}^K$.
1: **for** $k = 1, \ldots, K$ **do**
2:      Randomly initialize $\hat{\mathbf{v}}_{k,\texttt{init}}$ with unit norm;
3: **end for**
4: **for** $\ell = 1, \ldots, L$ **do**
5:      **for** $k = 1, \ldots, K$ **do**
6:          **if** $k \leq \ell$ **then**
7:              Receive $\mathbf{v}_{1,\ell-1}, \ldots, \mathbf{v}_{k-1,\ell-1}$
8:              $\mathbf{v}_{k,\ell,0} := \mathbf{v}_{k,\ell-1}$;
9:              **for** $t = 1, \ldots, T$ **do**
10:                  $\mathbf{g}_{k,\ell,t} := \Sigma\mathbf{v}_{k,\ell,t-1} - \sum_{k'=1}^{k-1} \frac{\mathbf{v}_{k',\ell-1}^\top \Sigma \mathbf{v}_{k,\ell,t-1}}{\mathbf{v}_{k',\ell-1}^\top \Sigma \mathbf{v}_{k',\ell-1}} \cdot \Sigma\mathbf{v}_{k',\ell-1}$;
11:                  $\mathbf{v}_{k,\ell,t} := (\mathbf{v}_{k,\ell,t-1} - \eta\mathbf{g}_{k,t,\ell}) / \|\mathbf{v}_{k,\ell,t-1} - \eta\mathbf{g}_{k,t,\ell}\|_2$
12:              **end for**
13:              Broadcast $\mathbf{v}_{k,\ell} := \mathbf{v}_{k,\ell,T}$
14:          **else**
15:              $\mathbf{v}_{k,\ell} := \hat{\mathbf{v}}_{k,\texttt{init}}$;
16:          **end if**
17:      **end for**
18: **end for**
19: **return** $\{\mathbf{v}_{k,L}\}_{k=1}^K$

---

**Algorithm 4** EigenGame-$\mu$

---

**Require:** $\Sigma \in \mathbb{R}^{d \times d}$; # of eigenvectors (workers) $K$; # of iterations $T$; global communication rounds $L \geq K$, step size $\eta$.
**Ensure:** Approximate eigenvectors $\{\mathbf{v}_k\}_{k=1}^K$.
1: **for** $k = 1, \ldots, K$ **do**
2:      Randomly initialize $\hat{\mathbf{v}}_{k,\texttt{init}}$ with unit norm;
3: **end for**
4: **for** $\ell = 1, \ldots, L$ **do**
5:      **for** $k = 1, \ldots, K$ **do**
6:          **if** $k \leq \ell$ **then**
7:              Receive $\mathbf{v}_{1,\ell-1}, \ldots, \mathbf{v}_{k-1,\ell-1}$
8:              $\mathbf{v}_{k,\ell,0} := \mathbf{v}_{k,\ell-1}$;
9:              **for** $t = 1, \ldots, T$ **do**
10:                  $\mathbf{g}_{k,\ell,t} := \Sigma\mathbf{v}_{k,\ell,t-1} - \sum_{k'=1}^{k-1} \mathbf{v}_{k',\ell-1}^\top \Sigma\mathbf{v}_{k,\ell,t-1} \cdot \mathbf{v}_{k',\ell-1}$;
11:                  $\mathbf{v}_{k,\ell,t} := (\mathbf{v}_{k,\ell,t-1} - \eta\mathbf{g}_{k,t,\ell}) / \|\mathbf{v}_{k,\ell,t-1} - \eta\mathbf{g}_{k,t,\ell}\|_2$
12:              **end for**
13:              Broadcast $\mathbf{v}_{k,\ell} := \mathbf{v}_{k,\ell,T}$
14:          **else**
15:              $\mathbf{v}_{k,\ell} := \hat{\mathbf{v}}_{k,\texttt{init}}$;
16:          **end if**
17:      **end for**
18: **end for**
19: **return** $\{\mathbf{v}_{k,L}\}_{k=1}^K$

---

