# OpenReview forum: "Efficient Distributed Principal Component Analysis with Parallel Deflation"
_ICLR.cc/2025/Conference — ICLR 2025 Conference Withdrawn Submission_

### Official Review · Reviewer_3eLv · 2024-10-30

**Soundness:** 4
**Presentation:** 4
**Contribution:** 1
**Rating:** 3
**Confidence:** 3

**Summary:**

This paper presents a model-parallel approach to PCA inspired by two previous papers by Gemp et al. More precisely, each agent is responsible for one principal component and optimizes it in a sequential manner. Convergence guarantees for such algorithm are provided for first time in the distributed setting. The authors also provide experiments in synthetic and real data.

**Strengths:**

The paper is very well-written. It presents a game-inspired approach for PCA developed in previous work. This game approach where the set of principal components is a Nash equilibrium is certainly interesting.

**Weaknesses:**

My objection with the paper is in a high level. I will phrase it as a question in the section below and I would like to see the authors' comment on it.

**Questions:**

My current understanding of the situation is the following:

The traditional subspace iteration $X_{t+1} = QR(\Sigma X_t)$, where $QR(\Sigma X_t)$ keeps the orthogonal factor of the $QR$ decomposition is naturally *data*-parallelisable. More precisely, if the dataset is split in batches, the covariance matrix $\Sigma$ is split in covariance matrices $\Sigma_i$ of smaller datasets, such that $\Sigma = \sum \Sigma_i$. Then, each agent holds a partial covariance matrix $\Sigma_i$ and the whole model approximation $X_t$. This is useful when the dataset is extremely large.

The approach developed in the previous Eigengame papers is model-parallelisable in the sense that each agent holds an approximation of one principal component (eigenvector) and the covariance matrix of the whole dataset $\Sigma$. This seems useful if the number of the wanted principal components is extremely large.

The problem is that in practice the number of principal components is never too large (max 30 in the experiments presented if I didn't miss something), while the number of data is often extensive (1.2 million in ImageNet). So, my question is why someone to want to parallelise the model of PCA in the first place? This question comes from a practical point of view and does not aim to diminish the theoretical importance of the Eigengame framework. I guess the two approaches (data and model parallelism) can be combined, but then still what is the added valued of model parallelism?

---

### Official Review · Reviewer_cxrc · 2024-11-01

**Soundness:** 3
**Presentation:** 4
**Contribution:** 3
**Rating:** 8
**Confidence:** 3

**Summary:**

The work "Efficient Distributed Principal Component Analysis with Parallel Deflation" proposes a novel algorithm to solve distributed PCA in the context where all agents have access to all data. As opposed to previously proposed methods this method allows agents (each agent in charge of finding an eigenvalue-eigenvector pair) to work in parallel without waiting for an accurate estimate computed by other agents.  Instead, agents start in a staged fashion as they wait for a first estimate of eigenvector-eigenvalue pairs computed prior to their own (descending order).

**Strengths:**

1)The paper provides convergence guarantees for decentralized PCA under the same setting as EigenGame but without requiring agents to sequentially find their corresponding eigenvalue-eigenvector pairs.
2) The paper's main technical theorem requires careful handling of the errors.
3) The end result, which is (near) linear convergence of each of the eigenvectors is what is expected.
4) the authors provide empirical evidence of the algorithm's performance, which yields similar performance as the decentralized variant of EigenGame for which no explicit guarantees exist.

**Weaknesses:**

Characterization of the main result: the main result is not easy to interpret. The rate for each eigenvalue is dependent not only on the performance of the top 1 operator for that specific eigenvalue, but also on the running average of all previous rates. Further the point at which this rate "kicks" in depends on the previous rates and also on consecutive eigenvalue gaps. While I understand that this is to a degree unavoidable, and it clear that the authors made an effort to make it more interpretable (see Remark 2), in particular, it would be good to have a connection between m_k and the eigenvalues of the sample covariance matrix.

**Questions:**

-

---

### Official Review · Reviewer_m9Yw · 2024-11-03

**Soundness:** 1
**Presentation:** 2
**Contribution:** 1
**Rating:** 1
**Confidence:** 5

**Summary:**

This work proposes a parallel algorithm for computing the k leading eigenvectors of a symmetric matrix, which is equivalent to solving PCA. The proposed algorithm is based on deflation with power iteration.

**Strengths:**

n/a

**Weaknesses:**

The major issue of this work is that it didn't survey the state-of-the-art classical numerical linear algebra methods for eigenvector computations, and the proposed algorithm based on deflation is not efficient. For computing the leading k eigenvectors of a symmetric matrix, there are existing packages for this problem, including parallel dense linear algebra packages such as ScaLAPACK, which is integrated with MKL that have routines for low-rank factorization, and sparse parallel libraries, such as PETSc, that have iterative solvers. Block Krylov methods are considered theoretically preferable for low-rank approximation, see for instance,

1. Gu M. Subspace iteration randomization and singular value problems. SIAM Journal on Scientific Computing. 2015;37(3):A1139-73.
2. Chapter 10 of "Matrix computations" by Golub and Van Loan.

I would suggest a rejection since the baselines above are not discussed in the paper.

**Questions:**

please see above

---

### Official Review · Reviewer_9TXD · 2024-11-03

**Soundness:** 2
**Presentation:** 3
**Contribution:** 3
**Rating:** 5
**Confidence:** 3

**Summary:**

This paper studies the Principal Component Analysis problem under a distributed framework where each eigenvector is computed in parallel by suitably aggregating the intermediate updates from the computations of all other eignevectors. Specifically, inspired by the EigenGame framework of Gemp et. al. (2020,2022), the paper presents a parallel deflation algorithm where each of the top K principal components are computed by K distinct workers in parallel. Each of the K workers starts with an initial estimate of the corresponding eigenvector, which is then iteratively refined in each round by a procedure where the kth worker uses updated estimates of the eigenvectors from the first k-1 workers, to update their eigenvector. This simulates the traditional deflation-based framework to find the top K eigenvectors, with the difference being that the kth worker need not wait till the first k-1 eignevectors have been computed exactly. The paper shows that, like EigneGame, the parallel deflation algorithm can be formulated as a game among K workers such that the true eigenvectors are the unique strict Nash Equilibrium of the game. The paper also provides a convergence analysis of the algorithm which shows that the kth eigenvector estimate converges almost linearly to the true kth eigenvector from round $s_k$ (where $s_k \geq S_{k-1}$ for all $k$) (as long as each of the workers in each round uses a subroutine that makes sufficient progress towards estimating the true eigenvector ). The paper also shows how the given algorithm can be used to estimate the principal components in a stochastic setting where, at each iteration, only a mini-batch of the data is used to estimate the covariance matrix. Finally, some experiments have been provided which show that the proposed algorithm has performance comparable to that of the EigenGame-based algorithms.

**Strengths:**

1) The algorithm presented in the paper is intuitive and basically parallelizes the usual deflation-based framework for computing the top K eigenvalues.
2) Though the idea is similar to the EigneGame framework, the paper presents a convergence analysis for the estimates of each of the top k eigenvectors.  Such an analysis is very non-trivial due to the fact that the deflation in each round is inexact and also, in the next round, the updates start using the eigenvector estimate from the previous round but uses the updated deflated matrix of the current round. This kind of analysis was missing from prior work like EigenGames. The experiments also show some promising results though there are some questions in this regard (see the weakness section).

**Weaknesses:**

Though the paper is interesting and the convergence analysis of the algorithm seems to be novel (I have not checked the detail of the proof) my major concern about the paper is regarding the experiments that have been performed.

1) One concern is the error metric used in the experiments. It seems that the error that is being computed is the sum of the squared $l_2$ error between the top $k$ true and approximate eigenvectors. I think this error might be misleading since in practice, we want to estimate the top eigenvectors corresponding to the larger eigenvalues more accurately. It may be the case that the error of the top eignevectors is pretty high but the aggregate error is small. A better error metric might be something like the "longest correct eigenvector streak" as measured by the angle between the estimated and true eigenvectors, which is what has been considered in the EigenGames paper. Or maybe just something like the angles between each of the the true and estimated eignevectors.

2) This is not a weakness per se, but I feel that in the "Our approach and contributions" section, the paper oversells itself a bit when it claims it presents a completely "Novel Algorithmic Framework". As shown in the experiments, the EigenGames algorithm can also be run in a distributed manner and converges to the true eigenvectors (albeit without a theoretical guarantee). In that case, I'm not sure if the claim that the paper presents a a completely new framework is fair.

To conclude, the major weakness of the paper is a lack of proper evaluation. Since this is primarily an ML conference, I feel that it is important to have proper experimental evaluation. I'm not sure if the paper warrants an acceptance based on just the theoretical contribution of Theorem 2.

Typo: In the abstract deflation is misspelled as 'delation' in line 4.

**Questions:**

1) How difficult will it be to extend the convergence analysis presented in the paper to EigenGames?
2) Does Theorem 2 hold for the case when there are repeated eigenvalues?

---

### Official Review · Reviewer_pzdD · 2024-11-04

**Soundness:** 4
**Presentation:** 4
**Contribution:** 3
**Rating:** 6
**Confidence:** 3

**Summary:**

proposes a novel method for distributed Principal Component Analysis (PCA), designed to enhance computational efficiency and scalability in large datasets. The authors introduce a Parallel Deflation algorithm, which diverges from traditional sequential deflation by enabling simultaneous, asynchronous updates from multiple workers, each targeting a distinct eigenvector.

**Strengths:**

1. By allowing parallel deflation without strict dependencies, this method addresses the synchronization bottleneck in distributed settings, an approach that enhances computational efficiency and could impact applications requiring real-time data processing.

2. The authors provide a robust theoretical foundation, supporting the algorithm’s convergence properties with detailed mathematical analysis.

3. Empirical results on benchmark datasets illustrate significant improvements over existing algorithms, particularly under high-dimensional data and asynchronous settings, underscoring the practical utility of the approach.

**Weaknesses:**

The analysis of the stochastic version Algorithm 2 is not included in this paper. Can we achieve similar theoretical guarantees in the stochastic setting?

**Questions:**

Please see the weakness.

---

### Note · Authors · 2024-11-26

I have read and agree with the venue's withdrawal policy on behalf of myself and my co-authors.